# Learning the Efficient Frontier

**Philippe Chatigny**[*]
Riskfuel
Toronto
pc@riskfuel.com

**Ivan Sergienko**[†]
Beacon Platform
New York
ivan.sergienko@beacon.io

**Ryan Ferguson**
Riskfuel
Toronto
rf@riskfuel.com

**Jordan Weir**
Riskfuel
Toronto
jw@riskfuel.com

**Maxime Bergeron**
Riskfuel
Toronto
mb@riskfuel.com

## Abstract

The efficient frontier (EF) is a fundamental resource allocation problem where one has to find an optimal portfolio maximizing a reward at a given level of risk. This optimal solution is traditionally found by solving a convex optimization problem. In this paper, we introduce NeuralEF: a fast neural approximation framework that robustly forecasts the result of the EF convex optimization problem with respect to heterogeneous linear constraints and variable number of optimization inputs. By reformulating an optimization problem as a sequence to sequence problem, we show that NeuralEF is a viable solution to accelerate large-scale simulation while handling discontinuous behavior.

## 1 Introduction

Making the least risky decision to maximize a cumulative reward over time is a central problem in machine learning and at the core of resource allocation optimization problems [26]. In modern portfolio theory [16], this problem is commonly referred to as the efficient frontier (EF) [36]. It involves distributing resources among $n$ risky assets to maximize return on investment while respecting various constraints. These constraints (e.g., maximum allocation allowable for an asset) are set to prevent aggressive or unrealistic allocations in practice. Finding the optimal solution to this optimization problem given the expected risk and return of each asset under a set of constraints can be done by solving a convex optimization problem of quadratic programs (QP) and second-order cone programs (SOCP). Although a single optimization problem is not time-consuming to solve, its current computational cost remains the most significant bottleneck when performing the simulations necessary for financial applications [22, 7].

Indeed, inputs to the EF problem such as future expectations of asset returns, co-variances, and even simulated client preferences on asset allocation are stochastic (c.f. Table 1). Thus, one needs to repeatedly solve the optimal allocation problem under a large number of different scenarios and Monte Carlo (MC) simulations [39] are commonly used to estimate the expected reward over time. Given an allocation function $g$ and stochastic input $Z$, we need to compute $\mathbb{E}(g(Z)) = \int g(z) f_Z(z) \, dz$ where $f_Z$ is the density function of $Z$. Sampling the empiric mean of $g(Z)$ approximates $\mathbb{E}(g(Z))$ with a convergence rate of $\mathcal{O}(1/\sqrt{N})$ where $N$ denotes

---

[*]Corresponding author

[†]Contribution was made when working at Riskfuel

37th Conference on Neural Information Processing Systems (NeurIPS 2023).

the number of samples. The computational cost of this simulation is therefore influenced by the size of $N$ and the cost of computing $g$. In spite of the vast literature of variance reduction techniques [5, 17] and the use of low-discrepancy sampling methods [10, 21] that aim to reduce $N$, the results of the simulation will be misleading if the assumptions are not valid, leaving the cost of running $g(z)$ as the principal bottleneck. This makes it practically impossible to run multiple MC simulations on different candidate assumptions in real time unless significant computational resources are available. Applications that heavily depend on MC simulations of the EF problem like basket option derivatives pricing face this bottleneck because their valuation depends on accurate estimates of the expected returns $\mathbb{E}(R)$, portfolio volatility $\mathbb{E}(V)$ and/or allocations $\mathbb{E}(X)$. The roll-out of new regulatory frameworks for financial applications of the EF problem (e.g. portfolio management) requiring more rigorous testing of $\mathbb{E}(g(Z))$ further increases the minimal acceptable $N$, exacerbating the need for speed [37, 3].

The core problem then becomes finding a way reduce the computational cost of $g$, making it possible to run a large number of MC simulations in a few seconds [27, 25]. Accelerating the optimization on highly-parallel hardware like graphical processing units (GPUs) can accelerate some convex optimization problems [13, 44, 11, 12], but not enough to run MC simulation in real-time effectively. Instead of only exploiting hardware, others have proposed to approximate the optimization step using deep neural networks (DNNs) and infer the result directly [18, 28, 48, 19]. However, proposed approaches fall short since they fail to simultaneously satisfy the following key requirements: (**1**) they do not provide theoretical guarantees that their forecast is within the domain set by constraints, (**2**) they do not show that their results can be applied at large scale faster than the optimization itself, (**3**) they can't handle a variable amount of heterogeneous inequality constraints and optimization inputs, and, (**4**) they are not robust to discontinuous behavior of optimization problems.

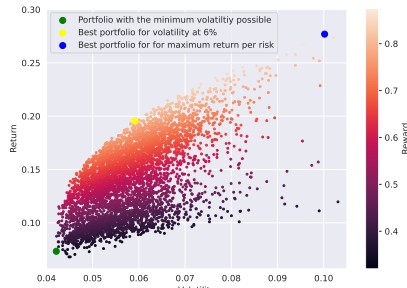

Figure 1: Illustration of the EF problem. The optimal allocations are at the frontier of the plot where the expected returns is maximized for a level of volatility. The yellow dot denotes the best allocation for eq. 2 for $\mathcal{V}_{\text{target}} = 0.06$. The location of the yellow dot will vary under different conditions, which can be approximated under different scenarios ($Z$) using MC simulations.

This work addresses all the shortcoming of DNN-based approaches mentioned above. We propose NeuralEF, a DNN-based model to approximate the computation of the EF problem robustly. Our model is up to 615X times faster than the baseline convex optimization when used with hardware acceleration. It is able to handle a variable number of assets and inequality constraints with consistent accuracy. In particular, we introduce a dynamic greedy allocation module to respect the constraints robustly that is agnostic to the DNN architecture. The remainder of this paper is organized as follows. Section 2 describes the EF problem and presents related works on convex optimization acceleration. Section 3 introduces NeuralEF and its training methodology. Section 4 outlines our empirical evaluations on the EF optimization problem. Finally, section 5 presents our conclusions.

## 2 Background and Related Work

### 2.1 The Efficient Frontier

We seek an optimal allocation $\boldsymbol{x} = [x_1, \cdots x_n]$ over $n$ *risky* assets given a soft volatility target $\mathcal{V}_{\text{target}}$. This means a portfolio with higher achieved volatility than $\mathcal{V}_{\text{target}}$ is valid only if the resulting allocation has the minimum volatility that can be achieved on the feasible domain. Finding the optimal allocation maximizing portfolio return at a given volatility target is conditional on the assets' expected returns $\boldsymbol{R} = [r_1, \cdots, r_n]$, volatilities $\boldsymbol{V} = [v_1, \cdots, v_n]$, and correlations $\boldsymbol{P} = [[\rho_{1,1}, \cdots, \rho_{1,n}], \cdots, [\rho_{n,1}, \cdots \rho_{n,n}]]$. Constraints that must be satisfied include upper and lower bounds on total allocation of *risky* assets $\alpha_{\text{MAX}}, \alpha_{\text{MIN}},$

and a minimum and maximum asset allocation limit per asset $\boldsymbol{X}_{\text{MIN}} = [x_1^{\text{MIN}}, \cdots, x_n^{\text{MIN}}]$, $\boldsymbol{X}_{\text{MAX}} = [x_1^{\text{MAX}}, \cdots, x_n^{\text{MAX}}]$. As is typical in finance, assets can belong to a set of $m$ classes $\boldsymbol{C} = [c_1, \cdots, c_n]; c_i \in [1, \cdots, m]$, and allocations are subject to a class maximum $\boldsymbol{\zeta}_{\text{MAX}} = [\zeta_{c_1}, \cdots \zeta_{c_m}]; \sum_{x \in c_j} x_i \leq \zeta_{c_j}$. We denote the set of convex optimization inputs by $\boldsymbol{Z}_{\text{input}} = [\boldsymbol{R}, \boldsymbol{V}, \boldsymbol{P}, \boldsymbol{X}_{\text{MAX}}, \boldsymbol{X}_{\text{MIN}}, \boldsymbol{C}, \boldsymbol{\zeta}_{\text{MAX}}, \alpha_{\text{MIN}}, \alpha_{\text{MAX}}, \mathcal{V}_{\text{target}}]$ and the resulting allocation by $\boldsymbol{Z}_{\text{output}} = \text{EF}(\boldsymbol{Z}_{\text{input}})$.

This problem is illustrated in fig. 1 using random allocations where the EF is located at the left-most frontier of the plots. We can find this frontier along a range of volatility targets by solving a two-step conditional convex optimization problem where we first find the optimal weights that minimizes risk in a portfolio of $n$ assets subject to linear weight constraints and then maximize the return if the risk of the portfolio is below $\mathcal{V}_{\text{target}}$.

Finding this minimum variance portfolio is equivalent to solving a QP problem of the form

$$\boldsymbol{\psi} := \text{minimize} \ \frac{1}{2}\boldsymbol{x}^\top \boldsymbol{Q}\boldsymbol{x} \ \text{subject to} \ \boldsymbol{a}_i^\top \leq \boldsymbol{b}_i \ \ \forall i \in 1, \cdots, w, \tag{1}$$

where $\boldsymbol{x} \in \mathbb{R}^n$ is a column vector of weights representing the allocation, $\boldsymbol{Q}$ is the covariance matrix of the portfolio's assets, $\boldsymbol{a}_i$ is a row vector representing the $i$-th linear constraint (obtained from $\alpha_{\text{MIN}}, \alpha_{\text{MAX}}, \boldsymbol{X}_{\text{MAX}}, \boldsymbol{X}_{\text{MIN}}, \boldsymbol{\zeta}_{\text{MAX}}$) and $\boldsymbol{b}_i$ is the maximum required value for the $i$-th constraint. The result of eq. 1 allows us to calculate the minimum volatility $\mathcal{V}_{\text{min}} = \boldsymbol{x}^\top \boldsymbol{Q}\boldsymbol{x}$ that can be achieved on the feasible domain.

If the resulting portfolio risk $\mathcal{V}_{\text{min}} < \mathcal{V}_{\text{target}}$, then we can afford to maximize portfolio return. In this case, the objective function of the first problem becomes one of the constraints of a SOCP problem of the form

$$\boldsymbol{\phi} := \text{minimize} \ -\boldsymbol{R}^\top \boldsymbol{x} \ \text{subject to} \ \frac{1}{2}\boldsymbol{x}^\top \boldsymbol{Q}\boldsymbol{x} \leq \mathcal{V}_{\text{target}} \ \text{and} \ \boldsymbol{a}_i^\top \leq \boldsymbol{b}_i \forall i \in 1, \cdots, w. \tag{2}$$

Thus, the efficient frontier can be summarized by

$$\boldsymbol{Z}_{\text{output}} = \text{EF}(\boldsymbol{Z}_{\text{input}}) = \boldsymbol{\psi} \ \text{if} \ \mathcal{V}_{\text{min}} > \mathcal{V}_{\text{target}} \ \text{else} \ \boldsymbol{\phi}. \tag{3}$$

The EF optimization is sensitive at inflection points where one asset becomes more attractive than another. All else held equal, an infinitesimal difference in expected returns can cause a jump in optimal allocation for all assets. Modeling the EF problem is challenging due to the presence of such discontinuities, which grows factorially ($\mathcal{O}(N!)$) as the number of assets increases.

## 2.2 Accelerating Convex Optimizations

The simplest way to speed up eq. 3 is to implement the solver for direct use with highly parallelizable hardware. This has been done using GPUs for various convex optimizations, particularly for large problems with numerous unknowns [13, 44, 12]. In particular, [11] showed that solving numerous optimizations simultaneously in a single batch leads to significant speedup. We replicated this experiment by writing a vectorized implementation of the optimization of eq. 3 in Pytorch to solve multiple EF problems at once. We observed accelerations which are consistent with the order of magnitude achieved in these works, but not sufficient to allow the completion of a full simulation in a matter of few seconds unless the GPU used has high memory capacity (e.g. 40 GB).

Several works [4, 1, 20] embed differentiable optimization problems in DNNs, offering a solution to robustly approximate the EF problem including the handling of the inequality constraints in eq. 1 and eq. 2. Their approaches aim to solve a parameterized optimization problem ensuring a constrained layer's output to align with homogeneous linear inequality constraints established by $\boldsymbol{Z}_{\text{input}}$. This is done either by solving the optimization problem during training [4, 1], or when initializing the model [20]. To support a heterogenous set of constraints, one would need to specify a set of linear inequality constraints ahead of time. This approach is impractical as it cannot generalize to unspecified sets of constraints.

Other existing supervised learning (SL) approaches to approximate convex optimization with DNNs do not provide the desired flexibility and robustness for large-scale generalization

[48, 2, 18, 28]. They use a DNN that takes $\boldsymbol{Z}_\text{input}$ as input and outputs $\boldsymbol{Z}_\text{output}$, treating inequality constraints as soft during training. Reinforcement learning (RL) approaches like POMO [31] and PPO [35] have also been applied variable length input combinatorial optimization problems with high label retrieval complexity where their learned policy tries to reduce an optimally gap by measuring the regrets from the optimal allocation. These DNN approaches do not guarantee that predicted values remain within the feasible domain and are often unable to handle the changing dimensionality of $\boldsymbol{Z}_\text{input}$ and $\boldsymbol{Z}_\text{output}$ based on $n$ and the set of constraints selected. Our proposed method addresses these issues, resulting in a fast and accurate approximation of the convex optimizer that respects constraints.

# 3 Neural Approximation of the Efficient Frontier

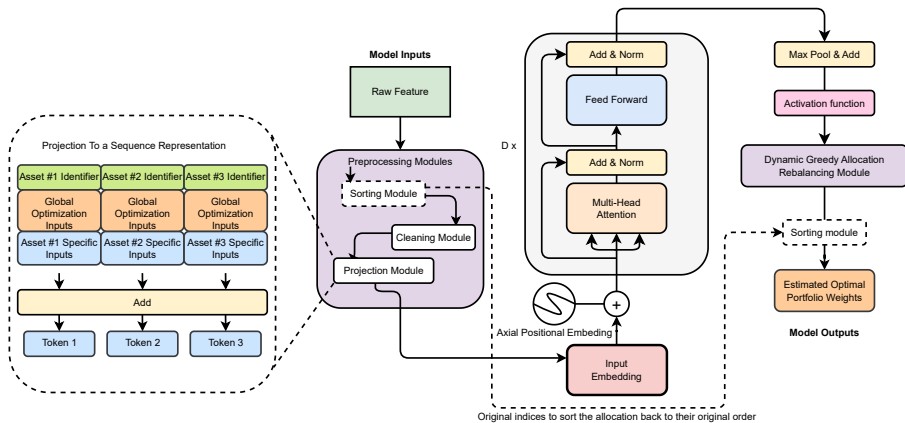

Figure 2: Illustration of the Transformer Encoder architecture of NeuralEF. On the left side the projection to a sequence representation from the optimization inputs considering 3 assets is shown as an example.

To approximate the computation of the QP and conditional SOCP optimization described in sec. 2.1, we use a stacked transformer encoder architecture [46, 29] as shown in fig. 2. The principal idea behind NeuralEF is to consider the optimization problem as a sequence-to-sequence (SEQ2SEQ) problem [45] and use the self attention mechanism to consider the relationships between the optimization inputs when approximating eq. 3. Contrary to large language models (LLMs) that solve quantitative mathematical problem by parsing the problem using a mix of natural language and mathematical notation where a forecast is built using the same notation [33, 15], our SEQ2SEQ formulation explicitly parse the whole optimization problem by considering the optimization input directly and forecasts the output in the solution domain. We setup the SEQ2SEQ problem such that the inputs are a set of tokens, each representing a single asset, and the outputs are a one dimensional sequence of the optimal allocation. To convert the

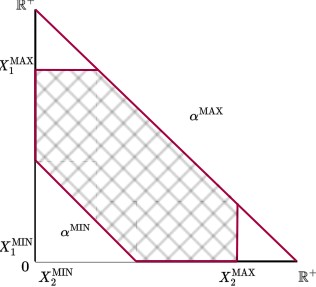

Figure 3: Illustration with two assets of the domain output space that the dynamic greedy allocation rebalancing module will enforce. The output domain is highlighted by the cross hatch patterns and the constraints are highlighted by the red lines.

optimization problem in eq. 3 to a sequence representation, we divide the optimization input parameters in two types of features: global optimization inputs ($\boldsymbol{P}, \boldsymbol{\zeta}_\text{MAX}, \alpha_\text{MIN}, \alpha_\text{MAX}, \mathcal{V}_\text{target}$) and asset specific inputs ($\boldsymbol{R}, \boldsymbol{V}, \boldsymbol{\Upsilon}_\text{MAX}, \boldsymbol{X}_\text{MIN}, \boldsymbol{C}$). These features along with a sequence of asset identifier ($n$ values evenly spaced within $[0, 1]$) are used to rearrange the optimization input parameters into a set of vectors and are linearly projected into higher dimensional

tokens $\boldsymbol{t}_1, \cdots, \boldsymbol{t}_n$ as shown in the left part of fig. 2[3]. Since the projection is made on numerical features instead of categorical embeddings like language model [9], the linear projection can express an infinity of tokens and this can induce learning difficulties during training.

Preprocessing optimization inputs before sequence projection is crucial to improving model performance and overcoming two training difficulties when considernig EF as a SEQ2SEQ problem. First, there are various input combinations that represent the same optimization problem. For example, when an asset's maximum weight surpasses its class constraint, the asset's maximum weight is effectively constrained by its class limit. We address this issue by preprocessing all inputs before being projected to remove any unnecessary complexity during training. Second, the model must be invariant to the sequence ordering [41, 49, 40] given as input. Building on insights from [47], NeuralEF sorts assets by returns, adding a positional embedding before the transformer encoder. This approach accelerates convergence to higher accuracy because assets with higher returns are more likely to receive greater allocation. When both of these approaches are combined, model training becomes significantly easier.

We introduce DGAR, a dynamic greedy allocation strategy, to constrain the encoder's forecast to stay within the domain set by the constraints, as shown in fig. 3. It consists of first clipping the forecast $X = [x_1, \cdots, x_N]$ with its upper and lower bounds $\boldsymbol{X}_{\max}$ and $\boldsymbol{X}_{\min}$ as in eq. 4. We extract an ordering of the assets $\mathcal{K} = [k, \cdots] := \mathrm{argsort}(-X; N)$ where $k \in [1, \cdots, N]$ such that assets with higher allocation are prioritized in the reallocation. We also compute an estimate of the total allocation based on the upper bound and lower bound of the domain using eq. 5. We then identify whether the total allocation is above $\alpha_{\mathrm{MAX}}$ by computing a cumulative sum following the ordering established previously and then altering the allocation on assets such that it does not exceed $\alpha_{\mathrm{MAX}}$ as in eq. 6. A similar process is applied to cases where $\tilde{\alpha}'_{\mathrm{ALLOC}} < \alpha_{\mathrm{MIN}}$, where we inflate the allocation of the assets greedily following $\mathcal{K}$ until we reach a total allocation $\tilde{\alpha}_{\mathrm{ALLOC}}$ which we compute by eqs. 7-10. The dynamic nature of the greedy allocation strategy allows using dynamic programming to perform the allocation in $O(\mathrm{Nlog(N)})$. DGAR guarantees that the forecast is within the allowed domain of linear constraints except for the class constraints $\boldsymbol{\zeta}_{\mathrm{MAX}}$ mentioned in sec. 2.1, and the volatility constraint (non-linear) which are both limitations of DGAR.

$$x'_i = min(max(x_i, x_i^{\mathrm{MIN}}), x_i^{\mathrm{MAX}}) \tag{4}$$

$$\tilde{\alpha}^{\mathrm{ALLOC}} = min(max(\sum_{i=1}^{N} x'_i, \alpha_{\mathrm{MIN}}), \alpha_{\mathrm{MAX}}) \tag{5}$$

$$x''_k = min\left(max\left(\begin{cases} x'_k + (\alpha_{\max} - \sum_{i=1}^{k} x_i)), & \text{if } \sum_{i=1}^{k} x_i > \alpha_{\max} \\ x'_k, & \text{otherwise} \end{cases}, 0\right), \alpha_{\mathrm{MAX}}\right) \tag{6}$$

$$m = \begin{cases} 1 & \text{if } \tilde{\alpha}'_{\mathrm{ALLOC}} < \alpha_{\mathrm{MIN}} \\ 0, & \text{otherwise} \end{cases}; \tilde{\alpha}'_{\mathrm{ALLOC}} = \sum_{i=1}^{N} x''_i \tag{7}$$

$$a = ||\tilde{\alpha}_{\mathrm{ALLOC}} - \tilde{\alpha}'_{\mathrm{ALLOC}}|| * m \tag{8} \qquad b_k = min(x_k^{\mathrm{MAX}} - x''_k, a) \tag{9}$$

$$x'''_k = x''_k + max(0, b_k - max(0, -(a - \sum_{i=1}^{k} b_k))) \tag{10}$$

## 4 Experiments

We used a MC sampling scheme to cover the entire domain in table. 1 uniformly to generate a dataset of approximately one billion samples $\mathcal{D}_{\mathrm{train}} = [(\boldsymbol{Z}_{\mathrm{input},1}, \mathrm{EF}((\boldsymbol{Z}_{\mathrm{input},1}), \cdots], $ which we used to train NeuralEF in a supervised fashion:

$$\theta_{\mathrm{NeuralEF}} = \underset{\theta^*_{\mathrm{NeuralEF}}}{\mathrm{argmin}} \frac{1}{N} \sum_{i=0}^{N} \mathcal{L}(\mathrm{NeuralEF}(\boldsymbol{Z}_{\mathrm{input},i}; \theta_{\mathrm{NeuralEF}}), \mathrm{EF}(\boldsymbol{Z}_{\mathrm{input},i})). \tag{11}$$

---

[3]There are many ways to set up an optimization problem into a SEQ2SEQ problem. Similar to prompt engineering on LLMs [8], different formulations influence the model accuracy. We do not focus on finding the best formulation to convert optimization problem into input tokens.

We generated two test datasets $\mathcal{D}_{\text{test}}, \mathcal{D}_{\text{validation}}$ of 1 million samples each on the the same domain as the training set described in table 1. All synthetic datasets mimic real-life distributions for the volatility and correlation inputs, encompassing area of the optimization domain around the discontinuity areas. Around 84% of the test samples feature at least two optimization inputs within $\epsilon$ proximity of each other to target discontinuity areas of eq. 3. This synthetic data allows for precise control over optimization inputs and encompasses a broad spectrum of scenarios, ranging from extreme to real-world cases targeting mainly the discontinuity regions. Consequently, it facilitates a comprehensive evaluation of the model's performance across various settings. All datasets have varying numbers of assets and asset classes, and include two allocation scenarios: full allocation ($\alpha_{\text{MAX}} = \alpha_{\text{MIN}} = 1$) and partial allocation ($\alpha_{\text{MAX}} > \alpha_{\text{MIN}}$) where there is no obligation to allocate the full budget. Only valid optimization inputs $\boldsymbol{Z}_{\text{input}}$, i.e. where $\text{EF}(\boldsymbol{Z}_{\text{input}})$ does not fail because of numerical difficulties in the solver or that no solution exists, where considered.

| Feature | Range | Feature | Range |
|---|---|---|---|
| ($\mathcal{V}_{\text{target}}$) volatility target | $[0.05, 0.15]$ | ($\boldsymbol{V}$) volatility | $[0, 2]$ |
| ($\boldsymbol{P}$) Correlation matrix | $[-1, 1]$ | ($\boldsymbol{R}$) returns | $[-1, 2]$ |
| ($\boldsymbol{\zeta}_{\text{MAX}}$) maximum class allocation | $[0.2, 1.0]$ | ($wt_{\text{MAX}}$) maximum asset allocations | $[0.01, 1.0]$ |
| ($\alpha_{\text{MIN}}$) Allocation lower bound | $[0.6, 1.0]$ | ($\alpha_{\text{MAX}}$) Allocation upper bound | $1.0$ |
| ($n$) Number of asset sampled | $[2, 12]$ | ($m$) Possible class | $[0, 1, 2]$ |

Table 1: Input Domain of optimization input used for training.

NeuralEF is a 7.9M parameter DNN and was trained on a single NVIDIA A100 GPU with stochastic gradient descent using the AdamW optimizer [34] and the $L^2$ loss. We also used an annealing learning rate decay starting from $5.5e^{-5}$ to $1.0e^{-6}$. As stated in sec. 2, we also implemented a baseline EF optimization in PyTorch that we used solely for comparing the evaluation throughput (evaluations/seconds) between NeuralEF over the base pricer which was implemented using CVXOPT [14]. The hyperparameters of NeuralEF are described in table. 2 and were selected by estimated guesses from the accuracy measured on $\mathcal{D}_{\text{validation}}$.

| Name | Value | Name | Value |
|---|---|---|---|
| Token dimension | 320 | (D) Transformer depth | 8 |
| Transformer # heads | 8 | Transformer heads dimension | 32 |
| Feed forward projection | 1024 | Output activation | Sigmoid |
| Embedding method | $[30, 24]$ | Hidden activation | Swish [42] |

Table 2: HyperParameters of NeuralEF. Adjusting accuracy at the expense of throughput can be easily done by increasing the token dimension.

## 4.1 In-domain Interpolation

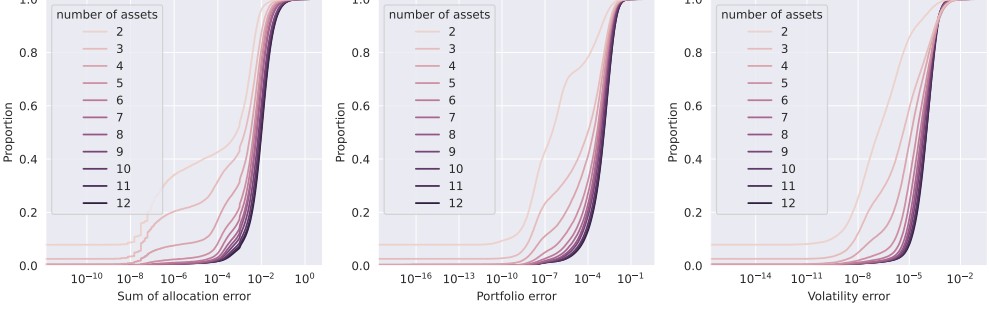

Figure 4: Cumulative distributions of the sum of absolute allocation error of allocations and portfolio returns per number of assets

We assess NeuralEF's accuracy by measuring its ability to predict portfolio weights, returns and volatility on $\mathcal{D}_{\text{test}}$. We use Mean square error (MSE), mean absolute error (MAE), and quantiles to evaluate the error distribution for portfolios with varying numbers of assets. Additionally, we test NeuralEF's precision in ranking asset importance in the allocation, check the precision of our model to forecast volatility ($\mathcal{V}_{\text{target}} = \boldsymbol{Z}_{output}^{\top} \boldsymbol{Q} \boldsymbol{Z}_{output} \leq max(\mathcal{V}_{\text{target}}, \mathcal{V}_{\min})$), and check for feasibility of violating constraints. Table 3 summarizes the accuracy results for different numbers of assets considered. Overall, NeuralEF shows accurate performance on portfolio weight prediction with slight deviations only in the higher upper quantiles for all asset cases. The model also demonstrates a high level of accuracy on returns and volatility. The ability to rank assets in order of importance correlates with the ability to respect constraints not captured by DGAR. The optimality gap metrics (ranking precision, $\boldsymbol{\zeta}_{\text{MAX}}$ and $\mathcal{V}_{\text{target}}$) are by design highly sensitive to a slight $\epsilon$ deviation, which in practice wouldn't necessarily be impactful. However, this limitation is to be considered when considering NeuralEF in safety-critical scenarios. NeuralEF can easily be applied within MC simulations that targets $\mathbb{E}(R)$, $\mathbb{E}(V)$, and/or $\mathbb{E}(x)$ as failure to respect the unsupported constraints does not necessarily lead to large error on the allocation. With a sufficiently large $N$, these errors do not impact the derived distributional features of $\mathbb{E}(g(Z))$.

| Asset case | Portfolio weights MSE | Portfolio weights MAE | 95 quantile | 99.865 quantile | 99.997 quantile | Ranking precision |
|---|---|---|---|---|---|---|
| 2 | 1.24e-06 | 1.11e-03 | 1.22e-02 | 3.39e-02 | 1.05e-01 | 91.122 % |
| 3 | 2.37e-15 | 3.97e-08 | 1.50e-02 | 3.78e-02 | 1.06e-01 | 99.080 % |
| 4 | 8.80e-07 | 6.03e-04 | 1.66e-02 | 4.57e-02 | 1.39e-01 | 98.421 % |
| 5 | 2.95e-06 | 1.33e-03 | 1.47e-02 | 4.15e-02 | 1.75e-01 | 96.587 % |
| 6 | 1.96e-05 | 2.02e-03 | 1.50e-02 | 4.35e-02 | 1.50e-01 | 93.896 % |
| 7 | 2.82e-06 | 1.19e-03 | 1.68e-02 | 4.43e-02 | 2.00e-01 | 90.426 % |
| 8 | 1.02e-05 | 2.35e-03 | 1.60e-02 | 4.59e-02 | 1.57e-01 | 87.861 % |
| 9 | 7.81e-06 | 1.67e-03 | 1.98e-02 | 5.17e-02 | 1.93e-01 | 84.895 % |
| 10 | 1.07e-05 | 2.15e-03 | 2.00e-02 | 5.23e-02 | 1.70e-01 | 81.519 % |
| 11 | 1.21e-05 | 1.78e-03 | 2.22e-02 | 5.83e-02 | 2.30e-01 | 80.003 % |
| 12 | 1.64e-08 | 1.04e-04 | 1.98e-02 | 5.28e-02 | 2.14e-01 | 78.067 % |
| | **Portfolio return MSE** | **Portfolio return MAE** | **95 quantile** | **99.865 quantile** | **99.997 quantile** | $\boldsymbol{\zeta}_{\text{MAX}}$ **precision** |
| 2 | 2.86e-06 | 1.69e-03 | 7.74e-03 | 2.20e-02 | 7.54e-02 | 100 % |
| 3 | 8.84e-06 | 2.97e-03 | 1.22e-02 | 3.15e-02 | 1.12e-01 | 93.927 % |
| 4 | 3.20e-06 | 1.79e-03 | 1.47e-02 | 4.10e-02 | 1.76e-01 | 90.221 % |
| 5 | 8.70e-06 | 2.95e-03 | 1.49e-02 | 3.60e-02 | 9.09e-02 | 89.406 % |
| 6 | 1.33e-10 | 1.15e-05 | 1.53e-02 | 3.63e-02 | 1.23e-01 | 88.690 % |
| 7 | 3.31e-10 | 1.82e-05 | 1.66e-02 | 3.94e-02 | 1.49e-01 | 85.186 % |
| 8 | 5.77e-06 | 2.40e-03 | 1.72e-02 | 4.22e-02 | 1.49e-01 | 83.326 % |
| 9 | 1.79e-06 | 1.34e-03 | 2.12e-02 | 5.18e-02 | 1.72e-01 | 87.125 % |
| 10 | 5.75e-10 | 2.40e-05 | 2.31e-02 | 5.33e-02 | 1.55e-01 | 83.003 % |
| 11 | 1.35e-07 | 3.67e-04 | 2.48e-02 | 5.80e-02 | 1.71e-01 | 81.934 % |
| 12 | 1.14e-06 | 1.07e-03 | 2.50e-02 | 5.72e-02 | 1.83e-01 | 83.999 % |
| | **Volatility return MSE** | **Volatility return MAE** | **95 quantile** | **99.865 quantile** | **99.997 quantile** | $\mathcal{V}_{\text{target}}$ **precision** |
| 2 | 2.86e-06 | 1.69e-03 | 7.74e-03 | 2.20e-02 | 7.54e-02 | 98.417 % |
| 3 | 8.84e-06 | 2.97e-03 | 1.22e-02 | 3.15e-02 | 1.12e-01 | 95.115 % |
| 4 | 3.20e-06 | 1.79e-03 | 1.47e-02 | 4.10e-02 | 1.76e-01 | 90.793 % |
| 5 | 8.70e-06 | 2.95e-03 | 1.49e-02 | 3.60e-02 | 9.09e-02 | 87.406 % |
| 6 | 1.33e-10 | 1.15e-05 | 1.53e-02 | 3.63e-02 | 1.23e-01 | 82.080 % |
| 7 | 3.31e-10 | 1.82e-05 | 1.66e-02 | 3.94e-02 | 1.49e-01 | 80.465 % |
| 8 | 5.77e-06 | 2.40e-03 | 1.72e-02 | 4.22e-02 | 1.49e-01 | 71.728 % |
| 9 | 1.79e-06 | 1.34e-03 | 2.12e-02 | 5.18e-02 | 1.72e-01 | 69.426 % |
| 10 | 5.75e-10 | 2.40e-05 | 2.31e-02 | 5.33e-02 | 1.55e-01 | 67.423 % |
| 11 | 1.35e-07 | 3.67e-04 | 2.48e-02 | 5.80e-02 | 1.71e-01 | 70.261 % |
| 12 | 1.14e-06 | 1.07e-03 | 2.50e-02 | 5.72e-02 | 1.83e-01 | 70.769 % |

Table 3: Accuracy of portfolio weights, implied return and resulting volatility

We plot the error distribution per asset by displaying the cumulative distribution of the sum error for all $x_1, \cdots, x_n$, the absolute error on the expected return and the portfolio volatility in Fig. 4. The total weight allocation error does not exceed 1% total deviation in around 98% of cases, but the impact on the resulting returns and volatility is only noticeable in 1% of cases. From the remaining tail exceeding the 1% total deviation, we measured no case violating the volatility constraint and only 8% exceeded the class constraint. The source of these errors was mainly found in regions with abrupt changes in the allocation due to the nature of the optimization.

NeuralEF accurately captures the qualitative behavior of the optimization in discontinuous regions, but instability in these areas exposes a weakness of approximating function with discontinuous behavior using DNNs. We illustrate this behavior using one example from the 99.997th quantile worst predictions of $\mathcal{D}_{\text{test}}$, where we vary a single input parameter ($r_{10} \in [-1.0, 2.0]$) while keeping the others constant, in Fig. 5. NeuralEF approximates discontinuities for most assets accurately, and errors on the portfolio returns are often attributed to ranking failure when NeuralEF transitions through these discontinuities. At the expense of always respecting constraints, DGAR can spread forecast errors to other assets with a poor $\mathcal{K}$ ordering leading to a larger absolute error in the resulting forecast than not using DGAR. Despite some assets not being modeled properly, we observe often that for practical purposes, this doesn't impact the resulting portfolio return or the portfolio

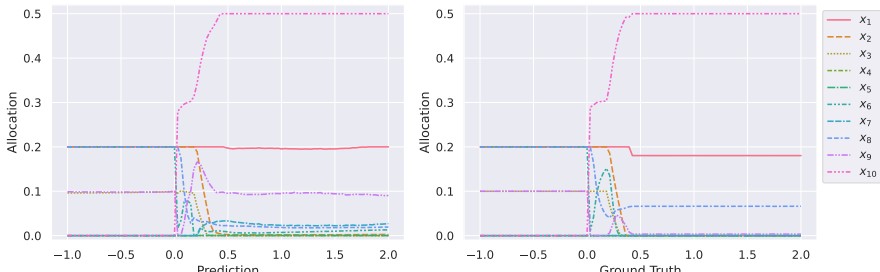

Figure 5: Illustration of the behavior NeuralEF at discontinuous points of the optimization. The unstable regions on the allocation occurs where $r_{10} \in [0, 0.5]$. All assets change allocation either smoothly through that region or suddenly by a jump discontinuity.

volatility, as illustrated in fig. 6 (a-b). An ablation study motivating the different NeuralEF's components is presented in sec. C.2

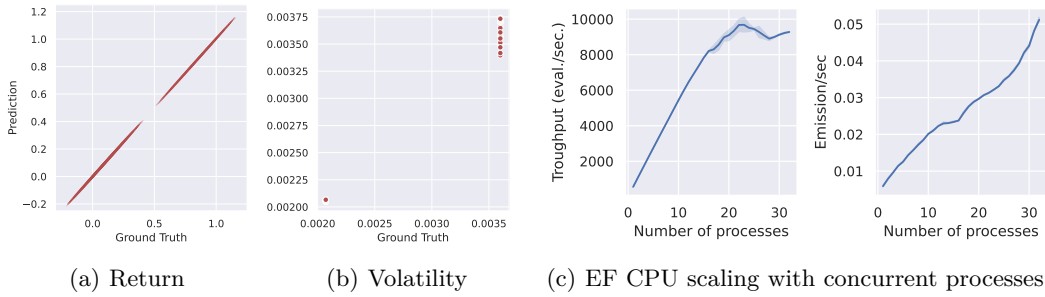

(a) Return       (b) Volatility       (c) EF CPU scaling with concurrent processes.

Figure 6: Impact of the inflection point prediction failure using the same prediction shown in fig. 5 on the estimated return and volatility respectively (a,b). EF numerical calculation throughput per concurrent process and its emissions in $kgCO_2e/kWh$ (c). Emissions presented are for computing 1000 evaluations per concurrent process and are scaled by a factor of 1e3.

## 4.2 Throughput Evaluation and Carbon Footprint Impact

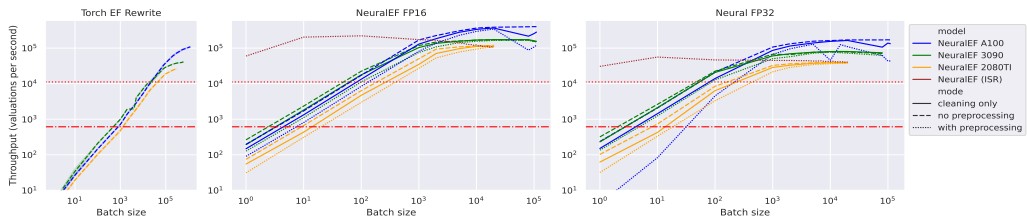

Figure 7: Throughput of the vectorized EF version on GPU (left), NeuralEF in FP16-precision (middle) and NeuralEF in FP32 precision (right) by batch size compared to the single-thread and multi-thread pricer.

We report a significant acceleration in the performance of NeuralEF compared to the single-tread baseline optimizer, with a 623X time improvement on a CPU (AMD 5950X achieving 559 eval./s). The main factors contributing to this improvement are running multiple optimizations in batches, the smaller computational cost of inferring the result compared to numerically solving the optimization using an interior-point solver, and the use of half-precision floating-point format[4]. We demonstrate the scalability of NeuralEF on fig. 7 for

---

[4]We don't observe a significant impact on accuracy using half-precision floating-point precision. See sec. C.3

both GPUs and CPUs as well as the impact of the different components. The highest achieved throughput are presented on table. 4.

Parallelizing the execution on CPU through concurrent processes of the base pricer can increase throughput, but the scaling ability of this approach is inferior to NeuralEF. Fig. 7 (c) shows that the throughput scales linearly with the number of cores until the concurrent processes saturate the CPU usage and prevent further speedup. Considering the AMD 5950X CPU as an example, we achieved a maximum throughput of 10377.80 eval./sec. [5]. It would take 26.77 hours to generate a billion evaluations, whereas it would only take 41 minutes or 1.25 hours with NeuralEF respectively on a A100 or on two ISR chips. When accelerated on GPU, impractical large batch sizes are needed to compete with a concurrent CPU setup. Pytorch-EF needs batches of 1.2 million request to reach a maximum throughput of 111479 eval./sec. In comparison, NeuralEF can achieves 343760 eval./sec. using batch size of 80k request. From an ease of use standpoint, the accelerated

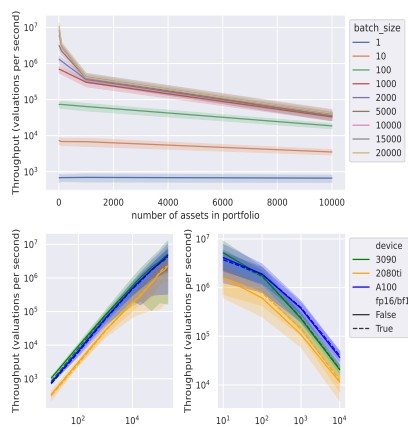

Figure 8: Scalability of DGAR throughput by number of assets and batch size.

throughput of NeuralEF on a few devices is more cost-effective than maintaining and running multiple machines to achieve the same throughput on CPU or having to deal with large batch size in practice. The scaling advantage of NeuralEF also offer an avenue to reduce the carbon footprint of large-scale simulations.

| A100: | Throughput (eval./sec.) | 2080TI: | Throughput (eval./sec.) |
|---|---|---|---|
| Pytorch-EF | 111479.39 | Pytorch-EF | 26452.06 |
| NeuralEF (fp32) | 128950.46 | NeuralEF (fp32) | 37821.05 |
| NeuralEF (fp16) | 343760.77 | NeuralEF (fp16) | 107259.29 |
| NeuralEF (fp16) clean-only | 366450.96 | NeuralEF (fp16) clean-only | 114160.65 |
| NeuralEF (fp16) no preprocessing | 401641.81 | NeuralEF (fp16) no preprocessing | 118711.91 |
| **Intel Xeon Platinum 8480+ (ISR)** | | **3090:** | **Throughput (eval./sec.)** |
| NeuralEF (fp32) | 56050.39 | Pytorch-EF | 41035.23 |
| NeuralEF (bf16 + AMX) | 221787.48 | NeuralEF (fp32) | 77859.95 |
| **AMD 5950X (reference)** | | NeuralEF (fp16) | 167594.15 |
| singe-thread | 559.15 | NeuralEF (fp16) clean-only | 173388.66 |
| Concurrent processes (23) | 10377.80 | NeuralEF (fp16) no preprocessing | 170650.62 |

Table 4: Maximum average throughput achieved. Note that two Intel Xeon Platinum 8480 CPUs were used simultaneously on a dual-socket machine to achieve the best throughput.

NeuralEF is less environmentally impactful than running the same optimization on CPU. For a single AMD Ryzen 9, if we consider the total time to generate our training dataset of 1 billion EF optimizations, the total simulation on CPU would approximate to 1.39 $kgCO_2e$. Increasing the throughput by running concurrent processes increases the $CO_2$ emission on the chip as it requires more energy as show in fig. 6 (c). Training NeuralEF is more environmentally expensive due to a cumulative 336 hours of computation on a single A100 GPU with an AMD server-grade CPU server resulting in approximately 5.71 $kgCO_2e$. At inference, the total emissions for forecasting a billion EF optimization using NeuralEF is estimated to be 0.03 $kgCO_2e$[6]. Hence, we can offset the total cost of training NeuralEF by running approximately 4.20 billions evaluations. Since MC error converges as $\mathcal{O}(1/\sqrt{N})$, one could get a 4x more precise approximation to $\mathbb{E}(g(Z))$ with NeuralEF in the time it would takes to obtain $N$ simulations on a single CPU. Given that the EF problem is a cornerstone

---

[5]CPUs of newer generations and with higher number of core are expected to perform better.

[6]$CO_2$ estimations were conducted using CODECARBON [43] for the throughput of NeuralEF and eq. 3. The "Machine Learning Impact calculator" [32] (MLIC) was used to estimate the cost of training. Both approaches were measured using a carbon efficiency of 0.025 $kgCO_2e$/kWh to approximate its use in an environmentally friendly data center. Since we didn't track total amount of our emission during training, the estimate for model emission made with the MLIC is less accurate.

of multiple applications in finance, 4.20 billions evaluations is easily offset in less than a week by a single organization.

## 4.3  Out-of-domain Generalization

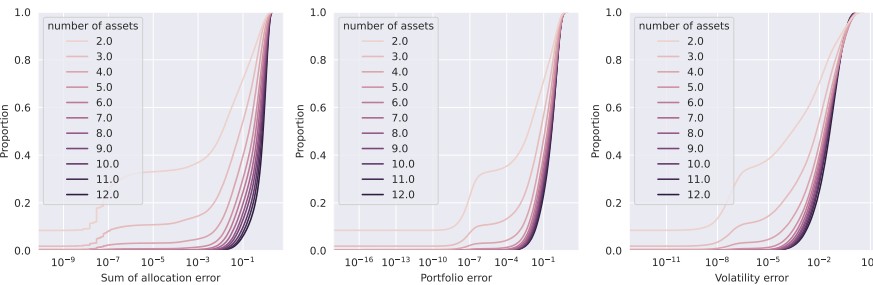

Figure 9: Cumulative distributions of the sum of absolute allocation error of allocations and portfolio returns per number of assets for $\mathcal{D}_{\text{ood}}$

The generalization ability of our model was measured by testing it on a larger input domain on a new dataset $\mathcal{D}_{\text{ood}}$ of 1 million samples using the same domain as in $\mathcal{D}_{\text{test}}$, but instead considering $\mathcal{V}_{\text{target}} \in [0.01, 0.3]$, $\boldsymbol{R} \in [0, 4]$, $\alpha_{\text{MIN}} \in [0.2, 1.0]$. The accuracy of the model degrades significantly on the tail of the distribution, especially for larger asset cases, indicating that it is not recommended to use NeuralEF outside the training domain. However, scale invariance in eq. 3 when all returns have the same sign (e.g., $\boldsymbol{R} = [0.3, 0.4, 0.5] \approx [3.3, 3.4, 3.5]$), allows the model to accurately estimate optimization even with returns outside the training domain by rescaling $\boldsymbol{R}$ by $\lambda$ such that they fit within the training domain. One can revert to the base optimization in the rare cases that it goes out of the domain, e.g. where assets go above 200% returns or the 200% volatilities, both being highly unlikely cases. Otherwise, one can always train NeuralEF on a bigger domain than table. 1 with most likely more data. Because NeuralEF has been trained on synthetic data, we haven't observed behavior bias that arises directly from certain regime aside in cases where assets would oscillate in the inflection area of the optimization, i.e. when two "attractive" assets have returns/volatilities $\epsilon$ close to each other and/or near 0 across a prolonged period of time.

## 5  Conclusion and Broader Impacts

In this work we introduce NeuralEF, a fast DNN-based model that approximates the solution of a convex optimization problem by treating it as a SEQ2SEQ problem. Using the Efficient Frontier, a highly discontinuous resource allocation problem, we demonstrate NeuralEF's accuracy and its ability to handle variable-length input optimization parameters while robustly respecting linear constraints by means of a novel dynamic greedy allocation rebalancing module. This positions NeuralEF as an attractive solution to reduce the computational footprint of running large scale simulations and offers a practical means to accelerate convex optimization problems in application that rely on MC simulation such as securities pricing, portfolio management and valuation of unit-linked insurance policies [6, 38].

More importantly, our reformulation of convex optimization as SEQ2SEQ offers a tangible step towards solving quantitative mathematical problems efficiently with DNNs [33, 15]. Whether it is easier to expedite optimization problems with heterogeneous constraints though a RL training paradigm [31, 35] or a SL paradigm like done with NeuralEF hinges on additional experimental insights. However, we know that current LLMs struggle to solve hard mathematical tasks [33] and simply scaling them is impractical for achieving strong mathematical reasoning [23]. Converting a quantitative mathematical problem into a sequence representation and then forecasting the results presents a more practical approach than solving it directly. This conceptually separate reasoning, i.e. how to formulate a mathematical problem given some inputs using a LLM, and solving it as two independent learning tasks that can be combined at inference.

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

# A  Numerical example of the EF problem

We provide a numerical example of the EF problem in the case of a 4-asset problem with 2 classes and $\alpha_{\text{MIN}} < 1$ (see Eq.12). Only the constraints are presented here. It is important to note that the matrices $\boldsymbol{A}$ and $\boldsymbol{B}$ representing all $w$ constraints grow in size by $(2n+2+m)$ as $n$ increases, where $m$ is the number of distinct asset classes. If $\boldsymbol{X}_{\text{MAX}}$ is not defined, then all entries of $\boldsymbol{X}_{\text{MAX}}$ can be set to $\alpha_{\text{MAX}}$. Solving Eq.1 is achieved by directly considering $\boldsymbol{A}$ and $\boldsymbol{B}$ and the covariance matrix $\boldsymbol{Q} = \text{diag}(\boldsymbol{V})\boldsymbol{P}\text{diag}(\boldsymbol{V})$ obtained from the other inputs. If $\mathcal{V}_{\text{min}} = \boldsymbol{x}^\top \boldsymbol{Q}\boldsymbol{x}$, then we solve a second-order cone program (SOCP) to increase the portfolio's volatility without exceeding $\mathcal{V}_{\text{target}}$ and get better returns.

$$
\boldsymbol{A} = \begin{bmatrix}
1 & 0 & 0 & 0 \\
0 & 1 & 0 & 0 \\
0 & 0 & 1 & 0 \\
0 & 0 & 0 & 1 \\
-1 & 0 & 0 & 0 \\
0 & -1 & 0 & 0 \\
0 & 0 & -1 & 0 \\
0 & 0 & 0 & -1 \\
-1 & -1 & -1 & -1 \\
1 & 1 & 1 & 1 \\
1 & 1 & 0 & 0 \\
0 & 0 & 1 & 1
\end{bmatrix}
\boldsymbol{B} = \begin{bmatrix}
0.591 \\
0.749 \\
0.412 \\
0.545 \\
0. \\
0. \\
0. \\
0. \\
-0.81 \\
1. \\
0.74 \\
0.58
\end{bmatrix} \quad (12)
$$

$$
\boldsymbol{C} = [0,0,1,1]; \boldsymbol{\zeta}_{\text{MAX}} = [0.74, 0.58]
$$
$$
\boldsymbol{X}_{\text{MAX}} = [0.591 0.749, 0.412, 0.545]
$$
$$
\boldsymbol{X}_{\text{MIN}} = [0,0,0,0]
$$
$$
\alpha_{\text{MIN}} = 0.81, \alpha_{\text{MAX}} = 1
$$

We obtain the volatility constraint through the Cholesky decomposition of the covariance matrix $\boldsymbol{Q}' = L(\boldsymbol{Q})L(\boldsymbol{Q})^T$ where $L$ is the lower-triangular operator. $\boldsymbol{E} \in \mathbb{R}^{[n+1,n]}$ is built by stacking $\boldsymbol{Q}'$ and $[0,\cdots 0]$ such that $\boldsymbol{F} \in \mathbb{R}^{[n+1,1]} = [\sqrt{\mathcal{V}_{\text{target}}}, 0, \cdots]$. Then, eq. 2 can be reformulated as follow:

$$
\phi := \text{minimize} \; -\boldsymbol{R}^\top \boldsymbol{x} \; \text{subject to} \; \boldsymbol{A} \le \boldsymbol{B} \; \text{and} \; \boldsymbol{E} \le \boldsymbol{F}. \quad (13)
$$

The complete optimal allocation of eq. 3 can be summarized by the following python script:

```python
"""EF evaluation """
import copy
import logging
import os

import cvxopt
import numpy as np

scalar = 10000

def cvxopt_solve_qp(P, q, G=None, h=None, **kwargs):
    P = 0.5 * (P + P.T)  # make sure P is symmetric
    args = [cvxopt.matrix(P), cvxopt.matrix(q)]
    if G is not None:
        args.extend([cvxopt.matrix(G), cvxopt.matrix(h)])
    sol = cvxopt.solvers.qp(*args, **kwargs)
    if sol["status"] != "optimal":
        raise ValueError("QP SOLVER: sol.status != 'optimal'")
    return np.array(sol["x"]).reshape((P.shape[1],)), num_iterations

def cvxopt_solve_socp(c, Gl, hl, Gq, hq, **kwargs):
    args = [cvxopt.matrix(c), cvxopt.matrix(Gl), cvxopt.matrix(hl), [cvxopt.matrix(Gq)],
    ↪    [cvxopt.matrix(hq)]]
    sol = cvxopt.solvers.socp(*args, **kwargs)
    num_iterations = sol["iterations"]
    if sol["status"] != "optimal":
        raise ValueError("SOCP SOLVER: sol.status != 'optimal'")
    return np.array(sol["x"]).reshape((Gl.shape[1],)), num_iterations

def efficient_frontier(max_weights, vol_target, conditions, condition_max, vol, ret, correl):
    n_assets = len(max_weights)
    min_weights = [0] * n_assets
    v_t = vol_target * vol_target * scalar
    G = np.vstack([np.eye(n_assets), -np.eye(n_assets), conditions])
    H = np.hstack([max_weights, min_weights, condition_max])
    cov = scalar * np.matmul(np.matmul(np.diag(vol), correl), np.diag(vol))
    wt = cvxopt_solve_qp(cov, np.zeros_like(ret), G, H) #eq 1
    wt = np.minimum(list(np.maximum(list(wt), min_weights)), max_weights)
    wt1d = wt.reshape([n_assets, 1])
```

```
variance = np.matmul(np.matmul(wt1d.T, cov), wt1d)[0, 0]
if variance < v_t:
    ret = np.array(ret)
    chol = np.linalg.cholesky(cov).T
    Gq = np.vstack([np.zeros(n_assets), chol])
    hq = np.zeros(n_assets + 1)
    hq[0] = np.sqrt(v_t)
    wt = cvxopt_solve_socp(-ret, Gl=G, hl=H, Gq=Gq, hq=hq) #eq 2
    wt = np.minimum(list(np.maximum(list(wt), min_weights)), max_weights)
return wt
```

## B    Preprocessing

We encounter ambiguity in optimization problems due to various combinations of inputs representing the same problem. To address this, we provide three examples where we discuss the ambiguity and propose a standardized solution for processing inputs in an optimized manner prior to token projection.

When the $i$-th asset belongs to the $j$-th asset class and $x_i^{\text{MAX}} > \zeta_{c_j}$, the constraint $x_i^{\text{MAX}}$ is overridden by $\zeta_{c_j}$. This means that there is no combination of assets where the allocation of the $i$-th asset can be higher than $\zeta_{c_j}$. To address this constraint, we clip $x_i^{\text{MAX}}$ to $\zeta_{c_j}$ by using the formula: $x_i'^{\text{MAX}} = \min(\max(x_i^{\text{MAX}}, \zeta_{c_j}), 0)$ for all $i$-th assets belonging to the $j$-th class.

The remaining two cases are additional edge cases related to the previous condition. If only one asset is assigned to the $j$-th class, $\zeta_{c_j}$ and $x_j^{\text{MAX}}$ should be equal because it is equivalent to having no class constraint for that class. Also, if a class constraint is set but no assets belong to that class, it is equivalent to setting $\zeta_{c_j} = 0$. By processing the optimization inputs in this manner, we ensure that any ambiguity on the class constraints are standardized, allowing for equivalent linear projections into token before the transformer encoder part of the network.

## C    Experimental Section

### C.1    Dataset

| Dataset name | Size | Description |
|---|---|---|
| $\mathcal{D}_{\text{train}}$ | 1.2B samples | Sampled at random over the domain in table. 1 |
| $\mathcal{D}_{\text{test}}$ | 990K samples | Sampled at random over the domain in table. 1 |
| $\mathcal{D}_{\text{ood}}$ | 990K samples | Sampled following the indication of sec. 4.3 |

Table 5: Description of the dataset used

The size and description of the dataset we used are presented in table. 5. We used an asymmetric weighting scheme to generate all datasets, favoring more complex optimizations (see Table 6). As the number of assets increases, the number of unstable regions also increases, where allocation can significantly change. To ensure training and evaluation encompass these unstable regions, we generated a higher proportion of optimization inputs with more assets.

### C.2    Ablation study

NeuralEF comprises three parts. First, the preprocesses step (1.1) sort requests by asset returns, (1.2) reduce optimization input ambiguity (Appendix B), and (1.3) project them into tokens for a SEQ2SEQ representation. Secondly there is a bidirectional encoder transformer that solve EF as SEQ2SEQ. Finally, there is DGAR to enforce constraints. The preprocessing and DGAR help accurate forecasting. (1.1) and (1.2) are optional for training if accounted in the training data but help at inference to ensure the sorting requirement of NeuralEF and clean request of some optimization input ambiguities.

To show the effectiveness of DGAR, we've trained two NeuralEF models (w/ and w/o DGAR) trained for 5 days with a fixed learning rate of $5.5e-5$ and present the distributional error an

| Asset case | Proportion (%) |
|:---:|:---:|
| 2 | 2.517 |
| 3 | 2.551 |
| 4 | 2.559 |
| 5 | 2.603 |
| 6 | 6.834 |
| 7 | 6.879 |
| 8 | 10.346 |
| 9 | 10.377 |
| 10 | 13.826 |
| 11 | 13.850 |
| 12 | 27.658 |

Table 6: Proportion of assets in each datasets.

 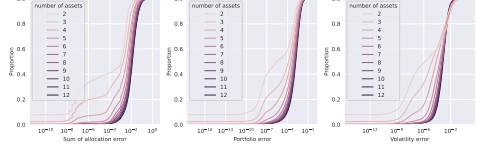

(a) $\mathcal{D}_{\text{test}}$ distribution error NeuralEF w/o DGAR trained w/ constant lr=5e-5 for 5 days

(b) $\mathcal{D}_{\text{test}}$ distribution error NeuralEF w/ DGAR trained w/ constant lr=5e-5 for 5 days

Figure 10: Accuracy W/ DGAR vs W/O DGAR

optimality gap of the metrics that are not respected on fig. 10-11 highlighting that model with DGAR achieves better MAE. Also, DGAR exhibits a reduced optimally gap of constraint adherence, further motivating the use of DGAR.

The cleaning module of NeuralEF were attached before the transformer part of the model to address the possible ambiguities provided in the optimization inputs. Using the same example of fig. 5 where the 10-th asset is associated to class $C_2$, which is its only member, we changed the value of the constraints to be ambiguous relative to another and show in fig. 12 how accuracy deteriorates when the model encounter request that are ambiguous.

## C.3 Accuracy with half-precision floating-point format

The results obtained using single-precision floating-point (FP32) and the model quantized to half-precision floating-point (FP16) are on the same order of accuracy as shown in table 7 and fig. 13. The quantization process to FP16 maintains the necessary precision for the calculations, resulting in equivalent outcomes as the FP32 counterpart. While a degradation

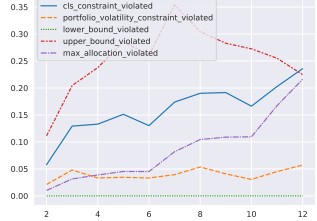 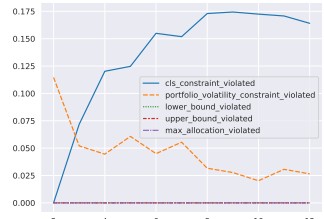

(a) $\mathcal{D}_{\text{test}}$ optimality gap NeuralEF trained w/o DGAR w/ constant lr=5e-5 for 5 days

(b) $\mathcal{D}_{\text{test}}$ optimality gap NeuralEF w/ DGAR trained w/ constant lr=5e-5 5 for 5 days

Figure 11: Optimiality gap W/ DGAR vs W/O DGAR

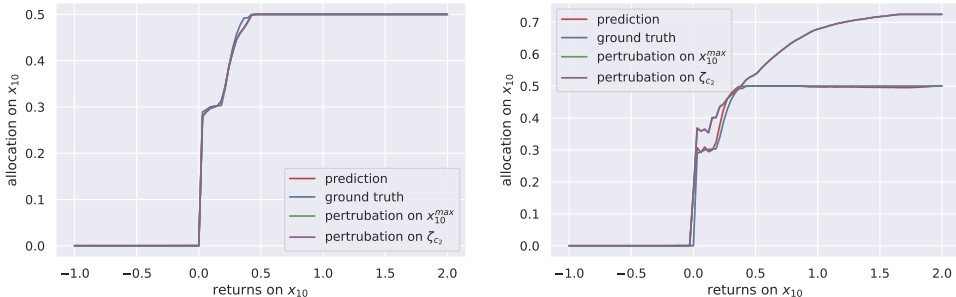

(a) Prediction of the 10-th asset **with** the cleaning module using the example of fig.6 under request perturbation

(b) Prediction of the 10-th asset **without** the cleaning module using the example of fig.6 under request perturbation

Figure 12: Issue of semantically equivalent requests with perturbation $x_{10}^{\mathrm{MAX}} \in [0.5, 1]$ and $\zeta_{c_2} \in [0.5, 1]$

in the ability to rank assets[7] and respect the volatility and class constraint occurs, we observe that this does not impact the overall distributional properties and the downstream applications that would benefit from it. As such, there is no discrepancy in the results between the two representations, demonstrating the viability of using the lower-precision FP16 for computational efficiency.

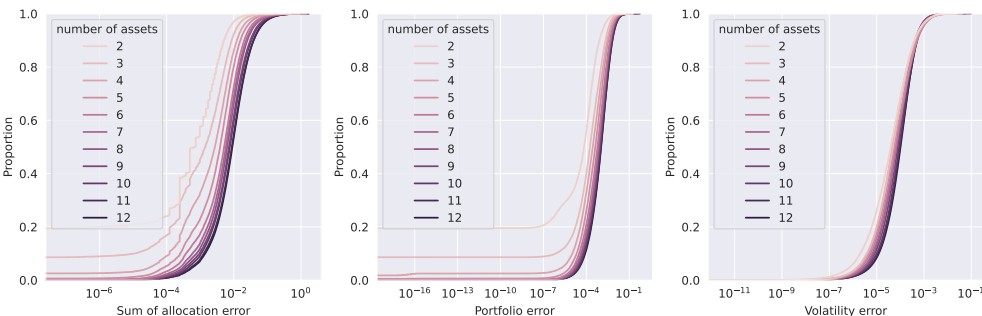

Figure 13: Cumulative distributions of the sum of absolute allocation error of allocations and portfolio returns per number of assets for the FP16 quantized NeuralEF.

---

[7]The ranking of the results has been computed with a tolerance of $1e-4$, where a slight deviations are permissible and don't hurt the accuracy. This was made such that negligible allocation made by NeuralEF which can be disregarded for practical purpose are neglected.

| Asset case | Portfolio weights MSE | Portfolio weights MAE | 95 quantile | 99.865 quantile | 99.997 quantile | Ranking precision |
|---|---|---|---|---|---|---|
| 2 | 9.54e-07 | 9.77e-04 | 1.25e-02 | 3.37e-02 | 1.09e-01 | 93.012 % |
| 3 | 7.95e-08 | 1.63e-04 | 1.51e-02 | 3.78e-02 | 1.06e-01 | 98.394 % |
| 4 | 6.85e-07 | 5.34e-04 | 1.65e-02 | 4.59e-02 | 1.39e-01 | 97.231 % |
| 5 | 2.52e-06 | 1.25e-03 | 1.48e-02 | 4.08e-02 | 1.77e-01 | 94.064 % |
| 6 | 1.94e-05 | 2.00e-03 | 1.51e-02 | 4.28e-02 | 1.68e-01 | 89.798 % |
| 7 | 2.84e-06 | 1.19e-03 | 1.67e-02 | 4.50e-02 | 1.95e-01 | 85.224 % |
| 8 | 1.06e-05 | 2.40e-03 | 1.61e-02 | 4.49e-02 | 1.57e-01 | 81.598 % |
| 9 | 7.72e-06 | 1.65e-03 | 1.99e-02 | 5.20e-02 | 2.08e-01 | 77.217 % |
| 10 | 1.16e-05 | 2.21e-03 | 2.00e-02 | 5.25e-02 | 1.71e-01 | 74.899 % |
| 11 | 1.11e-05 | 1.72e-03 | 2.22e-02 | 5.85e-02 | 2.36e-01 | 71.646 % |
| 12 | 3.48e-08 | 1.30e-04 | 1.98e-02 | 5.31e-02 | 2.11e-01 | 68.804 % |
| | Portfolio return MSE | Portfolio return MAE | 95 quantile | 99.865 quantile | 99.997 quantile | $\zeta_{\mathrm{MAX}}$ precision |
| 2 | 2.69e-06 | 1.64e-03 | 7.87e-03 | 2.26e-02 | 6.98e-02 | 97.555 % |
| 3 | 9.48e-06 | 3.08e-03 | 1.24e-02 | 3.19e-02 | 1.12e-01 | 90.949 % |
| 4 | 3.58e-06 | 1.89e-03 | 1.48e-02 | 4.05e-02 | 1.77e-01 | 87.576 % |
| 5 | 8.79e-06 | 2.96e-03 | 1.49e-02 | 3.57e-02 | 9.32e-02 | 86.583 % |
| 6 | 1.60e-13 | 3.99e-07 | 1.52e-02 | 3.58e-02 | 1.21e-01 | 85.592 % |
| 7 | 2.15e-09 | 4.63e-05 | 1.68e-02 | 3.94e-02 | 1.51e-01 | 82.323 % |
| 8 | 5.74e-06 | 2.40e-03 | 1.72e-02 | 4.25e-02 | 1.49e-01 | 84.808 % |
| 9 | 2.11e-06 | 1.45e-03 | 2.13e-02 | 5.19e-02 | 1.74e-01 | 83.589 % |
| 10 | 2.38e-09 | 4.88e-05 | 2.31e-02 | 5.36e-02 | 1.55e-01 | 83.707 % |
| 11 | 1.31e-07 | 3.62e-04 | 2.49e-02 | 5.75e-02 | 1.75e-01 | 83.190 % |
| 12 | 1.28e-06 | 1.13e-03 | 2.49e-02 | 5.75e-02 | 1.84e-01 | 83.197 % |
| | Volatility MSE | Volatility MAE | 95 quantile | 99.865 quantile | 99.997 quantile | $\mathcal{V}_{\mathrm{target}}$ precision |
| 2 | 2.69e-06 | 1.64e-03 | 7.87e-03 | 2.26e-02 | 6.98e-02 | 87.649 % |
| 3 | 9.48e-06 | 3.08e-03 | 1.24e-02 | 3.19e-02 | 1.12e-01 | 81.729 % |
| 4 | 3.58e-06 | 1.89e-03 | 1.48e-02 | 4.05e-02 | 1.77e-01 | 83.229 % |
| 5 | 8.79e-06 | 2.96e-03 | 1.49e-02 | 3.57e-02 | 9.32e-02 | 79.704 % |
| 6 | 1.60e-13 | 3.99e-07 | 1.52e-02 | 3.58e-02 | 1.21e-01 | 80.056 % |
| 7 | 2.15e-09 | 4.63e-05 | 1.68e-02 | 3.94e-02 | 1.51e-01 | 80.270 % |
| 8 | 5.74e-06 | 2.40e-03 | 1.72e-02 | 4.25e-02 | 1.49e-01 | 76.472 % |
| 9 | 2.11e-06 | 1.45e-03 | 2.13e-02 | 5.19e-02 | 1.74e-01 | 77.205 % |
| 10 | 2.38e-09 | 4.88e-05 | 2.31e-02 | 5.36e-02 | 1.55e-01 | 79.268 % |
| 11 | 1.31e-07 | 3.62e-04 | 2.49e-02 | 5.75e-02 | 1.75e-01 | 81.682 % |
| 12 | 1.28e-06 | 1.13e-03 | 2.49e-02 | 5.75e-02 | 1.84e-01 | 79.517 % |

Table 7: Accuracy of portfolio weights, implied return and resulting volatility for the FP16 quantized NeuralEF.

