# OpenReview forum: "Learning the Efficient Frontier"
_NeurIPS.cc/2023/Conference — NeurIPS 2023 poster_

### Official Review · Reviewer_o9F6 · 2023-07-04

**Soundness:** 2 fair
**Presentation:** 2 fair
**Contribution:** 2 fair
**Rating:** 5
**Confidence:** 2

**Summary:**

The authors propose a novel deep learning method, NeuralEF, to tackle large-scale constraint convex optimization problems by formulating them as sequence-to-sequence (SEQ2SEQ) problems. The key idea is to learn the complex relationship using an attention mechanism between financial conditions and optimal portfolio allocation, which is often a time-consuming task in portfolio management. They demonstrate the efficacy of NeuralEF on the Efficient Frontier problem in finance, a problem known for its non-continuous and highly variable nature. In the experiments, the authors utilize a synthetic dataset with varying conditions. The results illustrate NeuralEF's ability to effectively provide accurate portfolio allocations, all the while significantly reducing the computational resources required for large-scale simulations.

**Strengths:**

The originality and advancement of this paper lie in the proposed NeuralEF approach, which formulates a convex optimization problem as a SEQ2SEQ problem. The paper contributes to the development of the field by applying SEQ2SEQ modeling, originally a tool used in natural language processing, to a completely different domain of portfolio optimization. The concept of learning the mapping from financial conditions to portfolio allocation through deep learning has the potential to be applied to a variety of other optimization problems.


**Weaknesses:**

- This work seems to operate under the assumption that expected returns and volatilities are known with certainty. This might be a limiting factor when considering the application of this model in real-world scenarios, where such accurate estimation is challenging.
- The study lacks a rigorous evaluation on real-world financial data. While the synthetic data-based experiments might provide insights into the model's capabilities, the applicability of the model to real-world scenarios remains unclear without empirical validation on real datasets.
- It's also worth noting that modern portfolio approaches, such as risk-based portfolios, often rely less on expected returns due to the inherent difficulty in their estimation. Without comparative studies against such methods, it might be challenging to evaluate the practicality and superiority of NeuralEF.

**Questions:**

- Could the authors clarify how the NeuralEF model handles uncertainties in expected returns and volatilities? These are critical aspects in the field of asset management, and their treatment could significantly impact the model's performance and applicability.
- Could the authors consider evaluating the model using real-world financial data? Such evaluations could give more insight into how the model would perform in practice and help to assess its real-world applicability.
- How would the proposed model compare with other modern portfolio strategies, such as risk-based portfolios, that don't rely heavily on expected returns due to the difficulty in their accurate estimation?

**Limitations:**

- The paper does not address the assumptions made about the certainty of expected returns and volatilities. These assumptions could significantly impact the model's performance and applicability in real-world settings where such values are typically estimated with uncertainty.
- The practical utility of the proposed model remains somewhat uncertain without empirical validation using real-world financial data and without comparison with other portfolio approaches like risk-based portfolios, which are often used due to their reduced reliance on expected returns.

---

> ### Author Rebuttal · Authors · 2023-08-10
>
> # W(1) & Q(1):
> Both EQ(3) and NeuralEF rely on known expected returns, volatilities, and correlation matrices for optimization, yet they can be used in tandem with methods for handling uncertainty, such as Monte Carlo (MC). in the global author response, we provide two practical examples illustrating the relevance of NeuralEF with practical applications and how we can account input uncertainty. However, it's important to emphasize that our paper's focus is on accelerating EQ(3) solution. NeuralEF isn't a generalized framework for other portfolio-based methods like Minimum Variance Portfolio (EQ(1)), Risk Parity Portfolio, etc... Nonetheless, extending our work to encompass diverse portfolio allocation problems is intriguing, as these applications can be reformulated as convex optimization problems. NeuralEF effectively accelerates solving the Markowitz model [0], a cornerstone of modern portfolio theory with multiple financial applications. We provide two examples of application in the global author response in sec. "Example of real-life applications" and would be happy to mention one of the applications within introduction & related work of the manuscript in a revised version.
>
> # W(2) & Q(2):
> Assessing our model's performance on real financial data is useful for "what-if" scenario evaluations. We opted for synthetic data testing due to the following reasons: (1) Real-world financial data exhibit biases, conforming to established financial narratives, which undermines unbiased evaluation due to limited coverage of alternative financial events: if we sample two assets both in the same period, they all evolved with the same macro dynamics. (2) Variances in historical data length, correlation matrix computation, and volatility measure introduce complexities in allocation testing by altering optimization inputs, posing challenges for unbiased assessments of NeuralEF. Our synthetic data tackles these complexities by simplifying historical data selection and risk measures, while also encapsulating extreme real-world scenarios from [3-4]. It covers measures up to 200% volatility and return, providing a comprehensive outlook on model accuracy across all scenarios, including recent market crashes. We also target discontinuities by having 84% of the test data having optimization input near the discontinuity region. We can provide further details on these aspects in Section 4 with additional details about our synthetic data generation process as suggested in a different reviewer response.
>
> # W(3)& Q(3):
> as pointed out in W(1), NeuralEF is not meant to be a trading strategy or solve different equations than eq(3) by itself and we can point this out more clearly in the introduction and the abstract. A comparative study against different modern portfolio approaches on historical data for instance would allow us to answer what strategy works the best but this goes outside the scope of our paper. We provide a comparative study of NeuralEF against EQ(3) directly to assess whether the acceleration achieved by NeuralEF is sufficiently accurate which we showed in table 3 and figure 6. Additional details of this study are provided in the global author response within sec "Ablation study w/ and w/o DGAR" demonstrating its speed and accuracy advantages. The practicality and superiority of NeuralEF over running EQ(3) then becomes a question of (1) additional throughput, (2) ease of use of this methodology and (3) accessibility, (4) environmental cost of running the calculation.
>
> [0] https://en.wikipedia.org/wiki/Markowitz_model
> [1] Gatheral, Jim, and Antoine Jacquier. "Arbitrage-free SVI volatility surfaces." Quantitative Finance 14.1 (2014): 59-71.
> [3]: Mazur, Mieszko, Man Dang, and Miguel Vega. "COVID-19 and the march 2020 stock market crash. Evidence from S&P1500." Finance research letters 38 (2021): 101690.
> [4] Andersen, Torben G., et al. "The distribution of realized stock return volatility." Journal of financial economics 61.1 (2001): 43-76.

---

> > ### Comment · Reviewer_o9F6 · 2023-08-19
> >
> > Thank you to the authors for taking the time to address my questions and for providing additional experiments. I appreciate the effort and clarification provided. After reading all reviews and rebuttals, I would like to keep my score.

---

### Official Review · Reviewer_5SDP · 2023-07-05

**Soundness:** 3 good
**Presentation:** 3 good
**Contribution:** 2 fair
**Rating:** 5
**Confidence:** 4

**Summary:**

The paper presents a sequence-to-sequence (SEQ2SEQ) deep learning model, called NeuralEF, to solve the Efficient Frontier (EF) problem in portfolio optimization. The authors reformulate the EF problem as a SEQ2SEQ task and train a deep neural network (DNN) to approximate the convex optimizer. The proposed method aims to provide a fast and accurate approximation of the EF while handling variable-length inputs and robustly respecting linear constraints. A Dynamic Greedy Allocation Rebalancing (DGAR) module is introduced to ensure that the model's forecasts remain within the feasible domain and respect the constraints.

**Strengths:**

The paper proposes a novel deep learning-based approach to tackle the EF problem, leveraging the powerful capabilities of SEQ2SEQ models.
The DGAR module is an interesting addition that ensures the feasibility of the solutions generated by the DNN.

**Weaknesses:**

Lack of innovation in employing sequence-to-sequence models for convex optimization problems:
While the paper presents the NeuralEF method as a novel approach for solving convex optimization problems using sequence-to-sequence models, it is important to acknowledge that existing research has already explored similar ideas. For instance, sequence-to-sequence models have been applied to combinatorial optimization problems, and transformers have been used for embedding[1,2]. To strengthen the paper, the authors should discuss the novelty of their approach in light of these existing works and emphasize the unique contributions of their method to the field.


Inadequate demonstration of the effectiveness of the DGAR module:
The effectiveness of the Dynamic Greedy Allocation Rebalancing (DGAR) module, which is a key component of the NeuralEF method, is not thoroughly demonstrated or compared with the model's performance without it. This raises questions about the impact of the DGAR module on the model's performance and its ability to handle discontinuous behavior and diverse constraint types. To address this weakness, the authors should provide a more detailed analysis of the effectiveness of the DGAR module, including an ablation study or comparison with the model without DGAR.

References:
[1] Pomo: Policy optimization with multiple optima for reinforcement learning. Advances in Neural Information Processing Systems 33 (2020)
[2] Learning to iteratively solve routing problems with dual-aspect collaborative transformer. Advances in Neural Information Processing Systems 34 (2021)


**Questions:**

1 The formatting template used in this paper is incorrect; it should follow the NeurIPS 2023 guidelines instead.

2 Insufficient justification for the claimed speedup and practical significance:
The paper claims a significant speedup of the NeuralEF method compared to the baseline optimizer, but the justification for this speedup is not thoroughly provided. Furthermore, the practical significance of the speedup is not well addressed, considering the potential trade-offs between solution quality and training requirements. The authors claim that their NeuralEF method is 396 times faster than the baseline optimization method. However, if this speedup takes into account the 25x acceleration achieved by using a GPU, as mentioned in line 116, the actual speedup of the NeuralEF method compared to the baseline technique is approximately 15.84 times (396 / 25). It is important to consider whether this speedup is significant enough to justify the use of the proposed method over traditional techniques, given that this nework is already trained using 1 biliion samples. Moreover, there are concerns about the solution quality of the NeuralEF method, which might be worse than the baseline method when calculating the speedup.

3 Lack of comprehensive comparison with other optimization techniques:
 As shown in Table 3, the accuracy for Asset case 12 is below 84% across all three metrics. Without any comparison to other methods, it is difficult to determine if this level of accuracy is good or not. It also difficult to determine if it is sufficient for practical applications.

**Limitations:**

Inadequate demonstration of the effectiveness of the DGAR module:
The effectiveness of the Dynamic Greedy Allocation Rebalancing (DGAR) module, which is a key component of the NeuralEF method, is not thoroughly demonstrated or compared with the model's performance without it. This raises questions about the impact of the DGAR module on the model's performance and its ability to handle discontinuous behavior and diverse constraint types. To address this weakness, the authors should provide a more detailed analysis of the effectiveness of the DGAR module, including an ablation study or comparison with the model without DGAR.

Poor out-of-domain generalization:
The NeuralEF method exhibits poor performance when applied to out-of-domain examples, which limits its potential applicability in real-world scenarios.

---

> ### Author Rebuttal · Authors · 2023-08-10
>
> # W(1):
>
> [1-2] and our work explore input set optimization, each with distinct yet interconnected goals. Combinatorial optimization tackles discrete choices for optimal arrangements, while EF addresses continuous portfolio weight selection for risk-return equilibrium. PPO [2] and POMO [1], powerful RL algorithms, can potentially train NeuralEF for EF optimization as it can for problems where there exists a difficulty in retrieving training labels. However, unlike [1-2], the choice between RL and supervised training (ST) hinges on experimental insights due to the low label retrieval complexity of EF and complexity vs the NP-hard problem accounted for in these studies. POMO and PPO do not guarantee hard-constraint problem-solving, a void we bridge with DGAR. Uniting and evaluating these methods for convex optimization, like EF, is interesting so as to explore how to expedite optimization problems with heterogenous constraints, though the context favoring RL or ST remains unclear. We will acknowledge these nuances in our conclusion and references.
>
> # W(2) and limitations:
>
> We refer to our global author response in section "Ablation study w/ and w/o DGAR:" to address this point, where we further elaborate DGAR's effectiveness. On OOD generalization, training NeuralEF on a broader input domain beyond table 1 can encompass more extreme returns and volatility scenarios. For cases with assets having over 200% returns or volatilities (unlikely in reality), NeuralEF can revert to EF optimization. Hence, NeuralEF's OOD accuracy limitation applies only in such scenarios which considering [3-4] didn't occurs over multiple financial regime including recent financial crashes.
> # Q(1):
>
> We used the template on Overleaf authored by the NeurIPS Program Committee. If there are any formatting issues, please point them out explicitly, and we will gladly fix the errors.
> # Q(2):
>
> We revamped throughput experimentation, conducting ablation studies on: (1) preprocessing at inference under certain conditions, (2) DGAR's inference impact, (3) batch size vs. multi-threaded EF execution, and (4) comparisons with Torch-EF and NeuralEF*. Results validate NeuralEF's substantial speedup over the baseline optimizer. New achievable throughput data further confirms acceleration for both methods, offering practical trade-off insights. Including these results in the paper clarifies speedup metrics and prevents oversimplification of the X-fold improvement mention, allowing for a more nuanced understanding:
>
> a) Pytorch-EF achieves lower throughput than NeuralEF on an equivalent GPU, necessitating larger batches to surpass concurrent multi-thread CPU baseline throughput. Employing GPU-accelerated approach for many independent synchronized MC simulations is beneficial, though often infeasible due to insufficient parallel simulations. For instance, on an A100 GPU, Pytorch-EF requires 1.2 million optimization batches for 111479 evals/sec max throughput, whereas NeuralEF achieves 343760 eval/sec with 80k optimizations and 401641 eval/sec without cleaning requests.
>
> b) In fig.1 of the attached PDF, we observe the impact of preprocessing and DGAR on achievable throughput, even on relatively affordable hardware. Efficient EF computation can be further scaled using varied frameworks: CPUs optimized for matrix multiplication, like Intel Xeon Platinum 8480+, ideal for small requests (<1K), and GPUs for at least 1K requests, justifying the proposed method over traditional approaches. Capital derivative pricing, employing MC simulations of EF for single valuation, meets the GPU requirement with simulations often consuming substantial compute time. Accelerating them with minimal constraints (e.g., GPUs) yields significant time gains.
> # Q(3):
>
> Assessing constraints below the 84% level, not explicitly enforced in NeuralEF, is nuanced. In safety-critical scenarios demanding strict adherence, NeuralEF suits cases without ζmax requirement and Vtarget deviations. For MC simulations targeting E[R], E[V], and E[x], NeuralEF is suitable. These 3 metrics measure an optimality gap where NeuralEF predictions might not be the most "optimal" solution (similar to [1]). Ranking precision responds to ε deviations between assets, often with limited practical impact. To our knowledge, no other work handles robustly heterogeneous linear constraints for EF, making method comparisons a study of accuracy-constraint trade-offs. We address this trade-off in part in the global author response within sec "Ablation study w/ and w/o DGAR".
>
> [3]: Mazur, Mieszko, Man Dang, and Miguel Vega. "COVID-19 and the march 2020 stock market crash. Evidence from S&P1500." Finance research letters 38 (2021): 101690. [4] Andersen, Torben G., et al. "The distribution of realized stock return volatility." Journal of financial economics 61.1 (2001): 43-76.

---

> > ### Comment · Reviewer_5SDP · 2023-08-16
> >
> > Thank you for clarifying the effectiveness of the DGAR component and the novelty of your method. I find this discussion quite helpful, as it provides a clearer understanding of the paper's contributions.
> >
> >
> > Regarding the formatting template, I noticed that on the first page, the footnote indicates 2022. I believe the correct template should generate a PDF with a footnote displaying 2023.

---

> > > ### Author Response · Authors · 2023-08-16
> > >
> > > Thank you for your feedback. We're pleased to learn that the clarification on the DGAR component has contributed to a better understanding of our paper's contributions and the novelty of our approach. We believe that integrating the discussion and the figures from the rebuttal into the allowed additional page for the paper-ready version will strengthen the paper.
> > >
> > > Specifically, we propose to accentuate the following to provide an adequate demonstration of the effectiveness of the DGAR module and highlight the innovation of our approach for the EF convex optimization problems:
> > >
> > > [A]: Display the results from the model's ablation study exemplifying the impact of model component on throughput (Fig. 1), DGAR scalability (Fig. 2c), an example of discontinuities for volatility (Fig. 2b), and highlighting the effect of DGAR on accuracy by displaying the distributional accuracy of the model with and without DGAR either within the appendix or in the additional page if space allows.
> > >
> > > [B]: We will acknowledge the PPO [2] and POMO [1] methods, elucidating their relevance to our paper in both the related works section and the conclusion with discussion made in our answer.
> > >
> > > [C]: We will provide insights into the limitations of NeuralEF's OOD accuracy in practical scenarios, clarifying that such constraints are rare occurrences in practical scenarios as stated in our original response.
> > >
> > > [D]: We will enhance the discussion on the synthetic data generation in the experimental section, specifying that we employed an MC sampling scheme to uniformly cover the entire domain of table 1. Additionally, we will emphasize that our synthetic datasets mimic real-life distributions for volatility and correlation inputs, encompassing rare and extreme scenarios targeting optimization discontinuity areas (84% of the samples having at least 2 optimization inputs within ε proximity of each other). If space permits, we will further elaborate on the rationale for utilizing synthetic data to evaluate our model, as addressed in the response to reviewer "oNAD."
> > >
> > > [E]: We will comment on the optimality gap stated in section "Q(3)" of our answer to your review and elaborate on how our approach compares to the ground truth optimal result found with eq(3). We will highlight that NeuralEF is sufficient for practical applications in cases without strict requirement on ζmax and Vtarget deviations and is easily applicable in MC simulations settings targeting E[R], E[V], and E[x] based on the result from fig 4 in the paper. We will discuss the nuance of the optimality gap metrics of the volatility and the class constraints (both with respect to their sensitivity and strict use in practical applications) highlighting that these metrics are sensible to an ε deviation. To enhance this discussion, we can plot the distributional error on the total class allocation and for violating the constraint as a function of absolute error per number of assets and classes. We can do the same for the volatility constraint if you deem it to be required.
> > >
> > > [F]: We will provide a real-case application of NeuralEF highlighting a domain of application where our model would have an impact by using the first example stated in the global author response in sec "Example of real-life applications".
> > >
> > > [G]: We will comment further on the speedup of our method using the results shown in fig1 of the attached PDF to account for multiple references standpoint (GPU vectorized implementation, vs single-thread process vs multi-thread process) based on the point mentioned in section "Q(2)". This should allow for a more nuanced discussion in the experimental result that justifies the use of the proposed method instead of traditional techniques, i.e. in cases where multiple concurrent evaluation of eq(3) are needed.
> > >
> > > Regarding the formatting issue, we sincerely appreciate your keen eye for detail and for pointing out the formatting issue. We can correct this in a revised paper-ready version. Before writing this response, we have tested that our manuscript can be easily ported on the 2023 template (https://www.overleaf.com/latex/templates/neurips-2023/vstgtvjwgdng) and we confirm that it is compatible: our current manuscript is still limited to the nine content pages limit, including all figures and tables without our suggestion stated above.
> > >
> > > Please let us know if there are any other required additions to improve the score of the paper.

---

> > > > ### Comment · Reviewer_5SDP · 2023-08-17
> > > >
> > > > Thank you for the feedback. I will increase my score.

---

### Official Review · Reviewer_oNAD · 2023-07-07

**Soundness:** 3 good
**Presentation:** 3 good
**Contribution:** 3 good
**Rating:** 8
**Confidence:** 5

**Summary:**

This paper considers the problem of solving efficient frontier (EF), a fundamental resource allocation problem where one has to find an optimal portfolio maximizing a reward at a given level of risk. Traditionally, this optimization is solved by quadratic optimization techniques.

This paper introduces NeuralEF, a sequence-to-squence formulation, attempting to make fast neural approximations of EF. The purpose is to robustly forecast the result of the EF convex optimization problem with respect to heterogeneous linear constraints and variable number of optimization inputs.
By experimental results, the proposed  NeuralEF is a viable solution to accelerate large-scale simulation.

**Strengths:**

- The paper is well-written with clear notations and definitions.
- In the problem of solving for EF, obtaining robust and smooth results is a challenge given its discontinuous nature. The proposed method is shed light on how data-driven methods can help in smoothing the result in an organic way.
- r
- The methodology, especially the mathematical programming techniques, including formations in eq. 4-10 and the solver implementation details in Sec 4.2 are clear.



**Weaknesses:**

- Out-of-sample result is not discussed. It would be interesting to check the out-of-sample results in a distributional manner, to highlight the behavior bias of the proposed method in practical cases.
- Out-of-domain generalization is briefly discussed in Sec. 4.3. However, EF are dependent on correlation regime and shock, therefore it makes sense to discuss the solution accuracy in different regimes.

**Questions:**

- What behavior bias characteristics does the proposed NeuralEF have?
- In Sec. 4.1, what does the solution accuracy look like in different regimes?

**Limitations:**

no concern.

---

> ### Author Rebuttal · Authors · 2023-08-10
>
> # W(1-2)& Q(2):
>
> Out-of-sample results were not discussed in detail, mostly due to the page limit. The safe use of NeuralEF could be solved by training it on a bigger domain than what was shown in Table 1 (most likely on a longer time period and with more data) if the input domain needs to be larger. Otherwise, one can revert to the base optimization in the rare cases that it goes out of the domain, e.g. where assets go above 200% returns or the 200% volatilities, both being highly unlikely cases. With respect to correlation regime and shock, we can identify one area where NeuralEF accuracy would degrade significantly: in a regime where assets oscillate in the inflection area of the optimization, i.e. when two attractive assets have returns/volatilities  ε close to each other and/or near 0. If the returns are of the same sign and ε close to 0, then the scale invariance trick mentioned in 4.3 can be used to palliate to model potential forecast failures. Because NeuralEF has been trained on synthetic data, we haven't observed behavior bias that arises directly from certain regime aside from the behavior shown in fig 6.0, where there are non-continuous changes in allocation in EQ(3). We show on fig.4 c) the distributional error of the OOD data, similar to what we did for fig.4) in the paper. We can also measure the optimality gap of the constraint and provide them in the appendix of a revised version of the manuscript, if desired.
>
> # Q(1):
>
> Regarding DGAR’s behavioral characteristics, there is a possibility that DGAR propagates errors over other assets due to a bad selection of the K most important assets if it can do so with respect to the constraints. For instance, consider the case of three assets where the true allocation is x=[0.5, 0.3, 0.2], the estimated allocation before DGAR is x^=[0.2, 0.7, 0.3], the maximum assets allocation is x=[0.5, 0.6, 1.0], and we have a full allocation setup c1max=αmax=αmin=1 with all assets belonging to the same class. Then DGAR will clip the allocation at the first step x'=[0.5, 0.6, 0.2] and we will get the ordering K=[1,2,0], which will force the allocation to go back to x'''=x''=[0, 0.6, 0.4]. This allocation will have a higher MAE 0.33 vs the original prediction of 0.27. We haven't spotted areas of the input domain where there are more errors when forecasting x', but we can note that some of the bigger errors arise from this issue. Hence, it would be interesting to extend this approach in such a way that DGAR can account for the confidence it has over the allocation or learn an optimal ordering policy.

---

> > ### Comment · Reviewer_oNAD · 2023-08-20
> >
> > I thank the authors for their response. They have addressed my questions.
> >
> > I'd appreciate it if the authors can literally mention the discussed concerns and limitations (W(1-2), Q(2), Q1) in a final version.
> > It is optional to demonstrate the optimality gap of the constraint in the appendix of a revised version of the manuscript.
> >
> > I will maintain my rating.

---

### Official Review · Reviewer_Ygwb · 2023-07-22

**Soundness:** 2 fair
**Presentation:** 3 good
**Contribution:** 2 fair
**Rating:** 5
**Confidence:** 3

**Summary:**

In this paper, the authors propose NeuralEF, a deep neural network (DNN) approach that approximates what is known as the ``efficient frontier'' (EF) in economics. To this end, they use a stacked transformer encoder architecture, as well as pre-processing steps (such as ordering assets) and a greedy algorithm (DGAR) that uses dynamic programming to come up with an asset allocation. The main idea behind NeuralEF is to formulate the EF optimization problem as a sequence-to-sequence (SEQ2SEQ) problem. This allows them to use self attention inside a large language model (LLM) framework. The authors claim that this allows them to understand the relationships between the optimization inputs. They provide experimental results using NeuralEF on a convex optimization problem from generated data.

**Strengths:**

1. The paper is on an interesting topic: applying self-attention and LLMs to economics problems. This is important because many economic problems can not be solved in high dimension with standard tools and analysis from Economics.
2. Despite using a DNN, the authors are conscientious of their carbon footprint and compute resource usage. This makes their proposed approach more environmental as well as accessible to those with limited compute resources.
3. The authors wrote a vectorized implementation to solve multiple EF problems at once. Their proposed methodology is computationally efficient.
4. The authors augment existing DNN approaches by introducing separate modules that perform pre-processing and a dynamic greedy allocation module to respect constraints.

**Weaknesses:**

1. Some claims by the authors seem not to be substantiated. For example, they claim that their approach allows them to understand the relationships between the optimization inputs. Yet, there is little discussion of these relationships in the paper, and it is also unclear how their experimental results shed light in this area. The claim that their approach is robust to discontinuities resulting from large switches in assets appears also unsubstantiated.

2. NeuralEF has many moving parts and thus is difficult to understand and to use. It is unclear what the contribution of each component is. There are some details missing in the explanation of how NeuralEF works (e.g. how the initial allocation is chosen).

3. Beyond being able to leverage recent advances in LLMs, it is unclear why a SEQ2SEQ encoding of the problem makes sense in the context of solving the EF problem. Intuitively, multiple portfolios are equally efficient and there is no natural ordering of the assets in general.

4. The significance of the experimental results are unclear to me because the data generation process does not appear to be explained in the paper.

**Questions:**

1. What is an intuitive explanation of why it makes sense to sequentialize the assets? There is no natural ordering in a portfolio of assets and so, would you need to worry about making your solutions invariant to reordering?
2. What is the contribution of the main components of NeuralEF to the overall performance? For example, how much does the DGAR component affect the accuracy attained and the compute time needed? How much does the preprocessing contribute to the overall performance?
3. How does DGAR scale to larger number of assets? Even with $O(N\log N)$ complexity, this part seems costly with a sizeable portfolio.
4. Why include a class constraint when DGAR seems to just ignore it?
5. Does the result of NeuralEF depend on the initial asset allocation chosen?
6. How is the data generated? What is the function used to generate the data? What kind of noise is added to the data?
7. Why does the training set need to be so big (i.e. a billion examples)?
8. Why is the data generated for the experiments convex? Are there convexity assumptions or requirements for NeuralEF?
9. How does NeuralEF and the results in the paper contribute to understanding the relationship between optimization inputs to the EF problem?
10. The authors claim that discontinuities due to switching between assets doesn't impact the resulting portfolio return. Is this robustness to discontinuities also observed for the resulting volatility of the portfolio?

**Limitations:**

Yes

---

> ### Author Rebuttal · Authors · 2023-08-10
>
> # W(1)&Q(9):
>
> We meant to say that the use of self-attention is to help the model understand the relationships between inputs. The relationships of EF are well understood. An expansion in the appendix can detail these relationships: e.g. assets with higher expected returns and lower risks yield higher weights, leading to allocation shifts with small changes in returns or volatility around 0. The correlation matrix influences diversification benefits, shifting the efficient frontier. Constraints like maximum asset allocation impact the domain of the solution by truncating the attractivity of certain assets (as in Fig.3), etc. Regarding robustness in the presence of discontinuities, figures 5-6 (paper) and figure 2(b) of the global author response demonstrate a concrete example of our model in presence of the discontinuities. The test set was also designed to incorporate these discontinuities, with 84% of the samples having at least 2 optimization inputs within ε proximity of each other. Thus, we believe that table 3 of the paper accounts for the challenging optimization region because most of the test data samples are within the discontinuity region of the EF problem. We can point out this detail in the appendix.
>
> # W(2)&Q(2):
>
> We redirect the reviewer to the global author response in sec. "Ablation study w/ and w/o DGAR:" for answers to the reviewer’s questions and an explanation of the contribution of each part. We also provide an ablation study of the components motivating their contribution both in terms of accuracy and throughput.
>
> # W(3)&Q(1):
>
> Employing SEQ2SEQ to approximate EF holds a dual rationale: EF's optimization inputs grow by N (except the correlation, with has quadratic growth) and ordering optimization inputs on a standardized length basis (as illustrated in fig2 of the paper) allows to treat inputs as a set. Scaling-wise, solving a SEQ2SEQ problem is more resource-efficient than running the optimization itself. Since portfolio assets lack inherent ordering, any permutation of input tokens and output allocations is equivalent. Ensuring NeuralEF's invariance to reordering would enhance inference speed. However, we found that ordering tokens based on asset returns with [1.1] within a sequence accelerates training convergence and boosts accuracy, aligning with insights from https://arxiv.org/abs/1511.06391.
>
> # W(4)&Q(6-7):
>
> We used an MC sampling scheme to cover the entire domain (Table 1) uniformly. We made all synthetic datasets mirror real-life distributions for the volatility and correlation inputs, including rare and extreme scenarios like W(1), and explicitly target the area of the optimization where discontinuities are. We can present examples of the data generated in the appendix. Figure 2 of the attached PDF displays training set size impact on accuracy. We have observed that (1) smaller sets have lower forecast quality at inflection points and (2) larger sets enhance robustness in a wide variety of scenarios. Both small and large datasets can still achieve good MAE, but more data helps model robustness. Given the relatively low cost of generating training data we went with ~1B.
>
> # Q(3) & Q(4):
>
> DGAR doesn't encounter scaling problems, yet the number of inputs sent to GPU may be challenging. For instance, with 1K assets, dimensions to be sent reach a total of ~503k (mostly correlation inputs). Although there's no bottleneck in terms of complexity to solve the 1k portfolio under eq(3), DGAR's throughput varies with asset count (10, 100, 1000, 10000) and batch sizes (1 to 20000), as depicted in fig 2.c. Regarding constraints, we haven't found a way to make DGAR's respect both the maximum class and asset allocation constraints. To address this in the evaluation, we introduced metrics to measure if the class constraints and the volatility constraint were respected, allowing a direct comparison against the convex optimization eq(3), shedding light on our approach's optimality gap. We are happy to point out this detail in sec4.
>
> # Q(5):
>
> NeuralEF solves eq(3) without needing an initial allocation. Future work may consider an initial allocation and transaction costs with rebalancing constraints.
>
> # Q(10):
>
> Fig 2b) in the attached PDF illustrates the impact of inflection points, using the same example from fig.6 of the paper, where increasing the 10th asset's returns causes a jump in the allocation and affects resulting volatility. The optimal portfolio's volatility shifts with varying volatility levels, oscillating between 0.340% and 0.374% which is below the 95th quantile in term of accuracy.

---

> ### Comment · Reviewer_Ygwb · 2023-08-10
>
> Many thanks to the authors for their detailed response and additional experiments. I appreciate the clarification on why self-attention is needed. I will increase my score as I found the discussion and additional studies in the general response on DGAR and the preprocessing components helpful.

---

> > ### Author Response · Authors · 2023-08-16
> >
> > Thank you again for your thoughtful feedback and consideration of our response and additional experiments. Your decision to increase the score is greatly appreciated.

---

### Author Rebuttal · Authors · 2023-08-10

First, we would like to thank all reviewers for their appreciation of our paper and their valuable questions and comments. We answer all questions and provide new experimental results enhancing both the clarity of the paper and the experimental section. We would be happy to address all these aspects by stating the element of our responses in a revised paper-ready version and in the appendix to improve the paper.

# Summary of points and questions

We summarize the main points of our response from the questions and limitations stated in the review:

- Clarification on the EF problem, relationships between inputs.
- Clarification on the synthetic data generation and rationale for both training and test
- Relation with other RL works
- Ablation study of the NeuralEF components on accuracy and scaling and robustness on discontinuities
- Revamped throughput experimentation showing a more detailed picture of the speedup
- Optimality gap discussion & soundness of the experimental section w.r.t to accuracy of NeuralEF in real-case application and likelihood of in domain vs OOD accuracy.

Some of the questions and weaknesses mentioned express similar concerns. We've decided to discuss them in the global response to all reviewers to prevent repeating points across reviews. We explicitly mentioned in the official comment of the review whether a weakness (W) of a question (Q) is answered in this section.
# Ablation study w/ and w/o DGAR and preprocessing components:

NeuralEF comprises three parts. [1] preprocesses by [1.1] sorting requests by asset returns, [1.2] reducing optimization input ambiguity (Appendix B), and [1.3] projection for SEQ2SEQ. [2] a Bidirectional Encoder Representations from Transformers (BERT) that solve EF as SEQ2SEQ, with [3] DGAR to enforce constraints. [1] and [3] help accurate forecasting. [1.1] and [1.2] are optional for training if accounted in the training data but help at inference to ensure the sorting requirement of NeuralEF and clean request of some optimization input ambiguities. Fig.3 in the attached PDF illustrates an example of an ambiguity. We show an example where the 10-th asset is associated to class (c2), which is its only member. By changing the value of $\\zeta_{c_2}$ and $x\_1^{\\text{max}}$ constraints to be ambiguous relative to another, we show how accuracy deteriorates. This motivates the cleaning module. Figure 3.a-b) in the attached PDF displays the accuracy of NeuralEF models (w/ and w/o DGAR) trained for 5 days at a fixed learning rate of 5.5e−5, revealing better MAE with DGAR. DGAR exhibits a reduced optimality gap (Figure 4.a-b) of constraint adherence, further motivating the use of DGAR. Throughput of fig 1) shows that the cleaning steps are computationally inexpensive at inference.

Due to the page limit and time constraints associated to the author response period, a comprehensive ablation study considering various training data types (sort-returns/shuffled-assets;clean/no-clean) and usage of preprocessing components during training cannot be presented both for training and generating the data. This would requires us to generate 3Bs points and train multiple variants of the model for 5 days to be consistent with the other models presented in the ablation study. Note that all achieved throughput of fig 1) doesn't account for model optimization like the use of [A], which can increase the throughput of NeuralEF approach further.

# Example of real-life applications:

We provide concrete application in real-world scenarios where NeuralEF can be used:

Ex.1) Pricing financial derivatives tied to an asset basket that needs to estimate E[R], E[V], and E[x] from optimal portfolios to value options at different maturity times. Through Monte Carlo (MC) simulation using geometric Brownian motion, input uncertainty across time can be considered. Asset spot prices "S" can be simulated using the formula: Sᵢ₊₁ = Sᵢ * exp((interest_rateᵢ - (σᵢ^2)/2) * Δt + σᵢ * √(Δt) * Zᵢ, where Δt is the difference between ᵢ₊₁ and ᵢ, and Zᵢ is a normal distribution sample. This setup generates diverse paths under varying conditions, accommodating events like structural breaks in financial crashes. Calibration for volatility aspects can use option data and SSVI methods known in quantitative finance [1].

Ex.2) Quantifying forecast uncertainty impact on portfolio allocation: NeuralEF can be applied to historical data with multiple forecast degradation over the ground truth to see how error in the estimation of the optimization inputs will impact the portfolio performance over the most optimal allocation. This setup allows to back-test existing and proposed trading strategies reliant on expected returns and volatilities, highlighting suboptimal areas.

We are committed to incorporating the answer to your questions and comments into the manuscript. This will significantly strengthen the paper ensuring that we (1) highlight the novelty of NeuralEF, (2) show its impact on portfolio allocation settings and potential impact on other optimization problems with heterogenous constraints; and finally (3) make the evaluation convincing and accessible both in term of resources needed and reproducibility.

[A] https://dl.acm.org/doi/10.1145/3575693.3575702

---

### Decision · Program_Chairs · 2023-09-21

**Decision:**

Accept (poster)

**Comment:**

This paper proposes a neural-network based approach for computing approximate solutions of the efficient frontier problem. This is a portfolio optimization problem in finance where the goal is to compute an allocation of a small number of assets such that certain constraints (including volatility) are satisfied. The main contribution of this paper is the custom architecture for this problem and the empirical comparison of the approach in terms of solution accuracy and compute efficiency compared to a baseline solver.

The reviewers generally found the application of neural networks to solve the efficient frontier problem compelling and appreciated the presentation of this work. They also liked that the paper carefully compared the carbon footprint and compute resource usage of the proposed solution. However, there were also several concerns raised. Among them were:

* Missing ablation studies: The paper only presents an evaluation of the overall architecture but not which components are actually important.
* Missing comparisons to other neural-network approaches for approximately solving optimization problems such as *Pomo: Policy optimization with multiple optima for reinforcement learning.* or *Learning to iteratively solve routing problems with dual-aspect collaborative transformer.*
* Synthetic data generation and relation to real world applications: The paper does not disclose the data generation process and it is therefore not clear how close it is to real-world data.

The authors' rebuttal provided an ablation study of their preprocessing and DGAR module, key components of their approach. They also clarified the data generation process and the relationship to existing approaches and promised to include a more detailed discussion in the paper. All in all, the author's responses could alleviate the reviewers' concerns and every reviewer is leaning positively on this paper. The AC follows this sentiment and hence acceptance is recommended.